# IN-CONTEXT LEARNING FOR GAMES

## ABSTRACT

Most literature in algorithmic game theory focuses on equilibrium finding, particularly Nash Equilibrium (NE). However, computing NE typically involves repeated computations of best responses (e.g., policy space response oracle (PSRO)), which can be computationally intensive. Moreover, NE strategies may not be ideal in games with more than two players or when facing irrational opponents. Consequently, NE strategies often require further adaptions to effectively address various types of opponents, impeding practical deployments. In contrast, In-Context Learning (ICL), i.e., learning from context examples, plays the core role in the generalizability of large language models (LLMs) to novel tasks without changing parameters. While ICL has been applied to decision-making tasks, e.g., algorithm distillation (AD), existing research primarily focuses on single-agent scenarios, and the ICL for games is largely unexplored. To facilitate the game solving and the practical deployment, the research question investigated in this work is: *Can we leverage ICL to learn a model to i) play as **any player** of the game, ii) exploit **any opponent** to maximize the utility, and iii) be used to compute NE, **without changing the parameters**?* In this work, we propose **In-Context Exploiter** (**ICE**) to address this question: i) **ICE** generates the diverse opponents with different capability levels for each player of the game to generate the training datasets, ii) **ICE** combines the curriculum learning and the ICL for single-agent scenarios (e.g., AD), to train the single model for all players of games, and iii) **ICE** leverages the pre-trained single model to play as each player of the game against different opponents and integrate with the equilibrium finding framework, e.g., PSRO, to compute NE. Extensive experiments on Kuhn poker, Leduc poker, and Goofspiel demonstrate that **ICE** can efficiently exploit different opponents as different players of the games and can be seamlessly integrated with PSRO to compute NE without changing the parameters.

## 1 INTRODUCTION

Multiplayer games provide ideal testbeds of Artificial Intelligence (AI) research (Silver et al., 2018; Brown & Sandholm, 2019), aptly referred as "Drosophila of AI" (Omidshafiei et al., 2020). Most literature in algorithmic game theory mainly focuses on equilibrium finding and Nash Equilibrium (NE) (Nash, 1950) is the canonical solution concept for games, where no player can increase their utility by unilaterally deviating. There are two main issues impeding the practical deployments of NE into real-world scenarios. First, computing NE is typically computationally intensive, e.g., policy space response oracle (PSRO) (Lanctot et al., 2017) requires repeatedly computing best-responses, which is resource-demanding even for small-scale games, e.g., Kuhn poker (Kuhn, 1950). Second, NE is not always an ideal solution, particularly in games with more than two players or when facing irrational opponents, therefore, further adaptions of these strategies are needed to handle various opponents, which bring additional complexity and computational burdens to the deployment.

In-Context Learning (ICL), i.e., learning from context examples, is the core mechanism of the generalizability of large language models (LLMs) to novel tasks (Dong et al., 2022). Recent work (Laskin et al., 2022; Lee et al., 2024) extends ICL to decision making tasks, where a single policy can efficiently adapt to solve novel tasks from in-context examples, i.e., transitions. However, these works mainly focus on single-agent tasks and the ICL for games is largely unexplored. To facilitate the game solving and the practical deployment, in this work we provide a systematic investigation to ICL for games with the following core research question:

*Can we leverage ICL to learn a model to i) play as **any player** of the game, ii) exploit **any opponent** to maximize the utility, and iii) be used to compute NE, **without changing the parameters**?*

The three desiderata proposed in this paper aim to significantly improve the efficiency of equilibrium finding and enhance the practicability of the model for real-world deployment. However, there are several challenges to achieving these desiderata. First, we need to sample diverse opponents with different levels of capabilities against different players to generate the training sets, which is extremely important for the trained model to exploit the unknown opponents during inference. Second, the training process needs to be carefully designed, as more capable opponents are more difficult to exploit and the single model would suffer the catastrophic forgetting issues in exploiting various opponents. Third, how to integrate the model trained with ICL into the current equilibrium finding framework is still unclear, including both training and evaluation of the model. Therefore, novel methods are required to tackle ICL for games.

To address these issues, we propose the In-Context Exploiter (**ICE**). The main contributions of **ICE** include: i) **ICE** generates the diverse opponents for each player with the equilibrium finding algorithms, e.g., CFR (Zinkevich et al., 2007), and collects the trajectories of players as the training set, ii) **ICE** combines the curriculum learning and the ICL for single-agent scenarios, i.e., Algorithm Distillation (AD) (Laskin et al., 2022) and Decision-Pretrained Transformer (DPT) (Lee et al., 2024), to train the model without catastrophic forgetting, and iii) **ICE** leverages the trained model to play as each player of the game against any opponent and the trained model can also be used to compute NE with equilibrium finding algorithms, e.g., PSRO (Lanctot et al., 2017) during inference. Extensive experiments on Kuhn poker, Leduc poker, and Goofspiel demonstrate that **ICE** outperforms the NE strategies and RL methods, e.g., PPO, to efficiently exploit different opponents as different players of the games and can be used to compute NE without changing the parameters.

## 2 RELATED WORK

In this section, we provide a concise review of the related work of ICL for games. The first line of related work is opponent modeling, which basically uses the prior knowledge and the observations to infer the behaviors of an opponent (Nashed & Zilberstein, 2022). However, these methods usually are based on the explicit model of the opponent and update the belief of the opponents during the playing (Von Der Osten et al., 2017) or require further adaption, i.e., fine-tuning (Wu et al., 2022; Foerster et al., 2017), which brings additional complexities of the methods. Conceptually, ICL can be viewed as an implicit opponent modeling and only adapt the behaviors of the model through changing the in-contexts, which is much simpler than the current opponent modeling methods. The second line of related work is the equilibrium finding, which lies in the core research of game theory. CFR (Zinkevich et al., 2007) is a no-regret method, which plays the core role in the success on poker (Brown & Sandholm, 2019). Policy Space Response Oracle (PSRO) (Lanctot et al., 2017; Muller et al., 2020) is another popular framework for equilibrium finding, which starts with a restricted games with limited number of policies for players and iteratively adding new best-responses into consideration. Most equilibrium finding methods require the iterative computation of the best- or better-responses, which make these methods extremely computationally extensive. In this work, we intend to apply the ICL to facilitate the equilibrium finding. The third line of related work is in-context learning, which is the core mechanism of the remarkable generalizability of large language models (LLMs), e.g., GPT-4 (Achiam et al., 2023). By providing different in-context examples, LLMs can quickly generalize to novel tasks. Recent work (Laskin et al., 2022; Lee et al., 2024) successfully develop novel ICL methods to handle the decision making tasks with generalizability to novel tasks, which motivates us to investigate the ICL for games.

## 3 PRELIMINARIES

**Imperfect-Information Extensive-Form Games.** An imperfect-information extensive-form game (EFG) can be represented by a tuple $(N, H, A, P, \mathcal{I}, u)$ (Shoham & Leyton-Brown, 2008). $N$ is the set of players, i.e., $N = \{1, ..., n\}$ and $H$ is the set of histories which is the past action sequence. In particular, when the game starts, the history is an empty sequence $\emptyset$, representing the root node of the game tree. Additionally, every prefix of any sequence within $H$ is also included in $H$. There is a set of special histories, called terminal histories, which are sequences that end in the leaf nodes of the game tree. $Z$ is used to represent the set of terminal histories which is a subset of $H$, i.e., $Z \subset H$. $A(h) = \{a : (h, a) \in H\}$ is the set of available actions at any non-terminal history $h \in H \setminus Z$. $P$ is

the player function that maps each non-terminal history to a player, i.e., $P(h) \mapsto N \cup \{c\}$ in which $c$ denotes chance player, representing these stochastic events beyond players' control. The information set, represented by $\mathcal{I}_i$, forms a partition over the set of histories where $i$ takes action, such that player $i \in N$ cannot distinguish these histories within the same information set $I_i$. Therefore, each information set $I_i \in \mathcal{I}_i$ corresponds to one decision-making point for player $i$ which means that $P(h_1) = P(h_2)$ and $A(h_1) = A(h_2)$ for any $h_1, h_2 \in I_i$. For convenience, we can employ $A(I_i)$ and $P(I_i)$ to denote $A(h)$ and $P(h)$ for any history $h$ within $I_i$. $u_i$ represents the utility function of player $i$ that maps every terminal history to real numbers, i.e., $u_i : Z \mapsto \mathbb{R}$.

**Nash Equilibrium (NE).** The behavior strategy for player $i$, denoted by $\sigma_i$, is a function that maps every information set $I_i$ to a probability distribution over the available action $A(I_i)$. The set of strategies for player $i$ is denoted by $\Sigma_i$, i.e., $\sigma_i \in \Sigma_i$. Given a strategy profile $\sigma = (\sigma_1, \sigma_2, ..., \sigma_n)$, the expected value to player $i$ is the sum of the expected payoff of these resulting terminal nodes, i.e, $u_i(\sigma) = \sum_{z \in Z} \pi^\sigma(z) u_i(z)$. $\pi^\sigma(z) = \prod_{i \in N \cup \{c\}} \pi_i^\sigma(z)$ is the reaching probability of terminal history $z$ and $\pi_i^\sigma(z)$ is the contribution of player $i$ to reach the terminal history $z$. The common solution concept for the imperfect-information extensive-form game is Nash equilibrium (NE) (Nash, 1950), defined as a strategy profile such that no player can increase their expected utility by unilaterally switching to a different strategy. Formally, a strategy profile $\sigma^*$ forms an NE if it satisfies $u_i(\sigma^*) = \max_{\sigma_i' \in \Sigma_i} u_i(\sigma_i', \sigma_{-i}^*), \forall i \in N$, where $\sigma_{-i}^*$ refers to all the strategies in $\sigma$ except for $\sigma_i$.

**ICL for Decision Making.** AD (Laskin et al., 2022) collects the dataset of learning histories generated by a source RL algorithm and learns a causal transformer by autoregressively predicting actions given the cross-episode trajectory as context. The model trained by AD can be deployed to efficiently complete novel tasks. DPT (Lee et al., 2024) is another ICL method for decision making, where the context of DPT can be randomly sampled transitions and the prediction of the model is the optimal action of the query state. Both methods demonstrate the impressive ability of ICL to efficiently generalize to novel decision-making tasks without changing the parameters, which motivates us to extend the ICL to games to address the issues illustrated in the motivating example.

**Motivating Example.** Given the rock-paper-scissors (RPS) game, whose payoff table is depicted in Table 1, the only NE is $(\frac{1}{3}, \frac{1}{3}, \frac{1}{3})$ for both players and the expected utility of each player is 0. However, suppose that an opponent always plays rock, the expected utility of playing NE strategy against him is still 0, while the expected utility of always playing paper is 1. Therefore, playing the NE strategy would be a safe option but is not optimal in even two-player zero-sum symmetric games.

Table 1: Rock-Paper-Scissors

|   | R | P | S |
|---|---|---|---|
| R | (0, 0) | (-1, 1) | (1,-1) |
| P | (1, -1) | (0, 0) | (-1, 1) |
| S | (-1, 1) | (1, -1) | (0, 0) |

Simplex-NeuPL (Liu et al., 2022) also highlights a similar motivation for deviating from NE strategies in certain scenarios. Furthermore, computing NE usually requires the iterative computation of best-responses where each iteration in PSRO (Lanctot et al., 2017) is using RL methods, e.g., PPO, to train the policy from scratch for each player, which is extremely time-consuming even for small-scale games, e.g., Kuhn poker. The inefficiency of the equilibrium finding algorithms and the impracticability of the NE strategies for deployment motivates us to investigate the ICL for games to leverage one pre-trained model to i) play as any player, ii) exploit any opponent, and iii) be used to compute to the NE without changing the parameters.

## 4 IN-CONTEXT EXPLOITER (ICE)

In this section, we introduce In-Context Exploiter (**ICE**), specifically crafted to train a model to exploit any unknown opponent and increase its utility through in-context learning. Fig. 1 provides an overview of **ICE**, which includes three stages: i) collecting interactive histories via any RL algorithm with diverse opponent strategies, ii) training a model within a curriculum learning framework, and iii) employing the trained model to play as any player in opponent exploitation and computing equilibrium in equilibrium finding algorithms. Each of these stages plays a critical role in ensuring the model's adaptability and generalizability.

### 4.1 STAGE I: DATA COLLECTION

**Opponent Strategies Generation.** Let $\mathcal{D}$ denotes the set of opponent strategies. The key is that $\mathcal{D}$ should consist of diverse and representative opponent strategies. To this end, we employ two methods: *random generation* and *learning-based generation*, and the corresponding sets are respectively

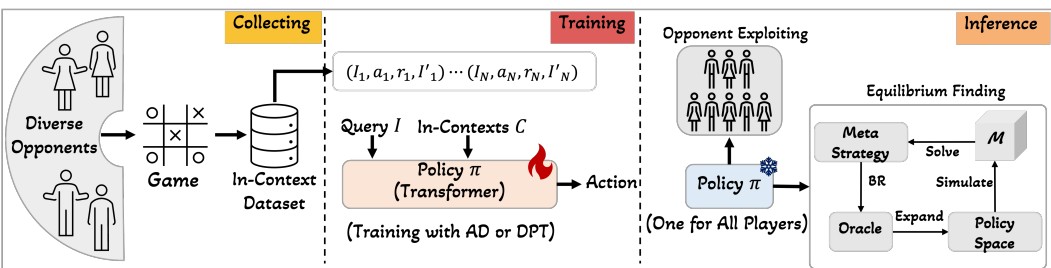

Figure 1: The three stages of ICE

denoted as $\mathcal{D}_r$ and $\mathcal{D}_l$. Suppose $\mathcal{D}_r = \{\mathcal{D}_{r1}, \cdots, \mathcal{D}_{rm}\}$ which includes $m$ random opponent strategies. Specifically, for each opponent's possible information set, we randomly generate a probability distribution over the set of available actions. This randomness ensures that the generated opponent strategies are highly diverse and include a variety of unpredictable opponent strategies, mimicking scenarios where opponents may act irrationally. On the other hand, let $\mathcal{D}_l = \{\mathcal{D}_{l1}, \cdots, \mathcal{D}_{lb}\}$ which consists of $b$ opponent strategies generated via the learning-based method. We take CFR (Zinkevich et al., 2007) as an example. Specifically, we run CFR for $n$ iterations where at each iteration, the opponent's average strategy among all previous strategies is added to the set $\mathcal{D}_l$. Intuitively, with the progress of the CFR process, the generated average strategies will approach the NE strategy. From the definition of NE, $\mathcal{D}_{l1}$ and $\mathcal{D}_{lb}$ are respectively the easiest and hardest strategies to exploit, which shows that $\mathcal{D}_l$ includes diverse opponent strategies spanning a range of skill levels.

**Interactive History Collection.** With a slight abuse of notation, we use $\mathcal{D}_{ri}$, $1 \leq i \leq m$, (resp., $\mathcal{D}_{li}, 1 \leq i \leq b$) to denote the in-context dataset corresponding to the opponent's randomly generated strategy $\mathcal{D}_{ri}$ (resp., learning-based generated strategy $\mathcal{D}_{li}$) as it is clear from the context. Following AD (Laskin et al., 2022), we can utilize any RL algorithm to play against the opponent strategy $\mathcal{D}_{ri}$ (resp., $\mathcal{D}_{li}$) and record the corresponding learning trajectories, each of which consists of the sequence of information sets, actions taken, and the resulting utilities: $(I_0, a_0, r_0, \cdots, I_T, a_T, r_T)$. In our experiments, we employ proximal policy optimization (PPO) (Schulman et al., 2017) to collect the trajectories. For DPT (Lee et al., 2024), the in-context training dataset consists of three components: in-context data (i.e., interactive history data), query states, and optimal actions. Unlike AD, DPT requires an optimal strategy to give the optimal action for the corresponding query state. For this purpose, we can leverage the strategy learned via PPO as mentioned previously as the optimal strategy when collecting in-context datasets for DPT.

## 4.2 Stage II: Training

Recall that we aim to train a Transformer model such that it can play as each player of a game against any opponent and can be used to compute NE with equilibrium finding algorithms during inference. The most straightforward idea is to directly train the model on the collected datasets $\mathcal{D} = \mathcal{D}_r \cup \mathcal{D}_l$, which could be unstable and inefficient since the model must adapt to a wide range of behaviors and tactics represented by these datasets. To overcome this issue, we devise a novel curriculum training framework to stabilize the training process and improve the training efficiency.

**Curriculum Generation.** As mentioned in the previous section, for the learning-based generated opponent strategies, $\mathcal{D}_{l1}$ is the easiest task while $\mathcal{D}_{lb}$ is the most difficult task. For the randomly generated opponent strategies $\mathcal{D}_r$, while it is possible to sort them based on the gap between these strategies and the NE strategy, computing the gaps would be time-consuming due to the large number of opponent strategies. Nevertheless, we observe that most of the randomly generated opponent strategies tend to be relatively simple to exploit, i.e., simple tasks. Therefore, we intersperse them into the learning-based generated opponent strategies to generate an effective curriculum $\mathcal{D}_o$, which is shown in Algorithm 1. Specifically, for every $g$ learning-based generated opponent strategies, we add a random opponent strategy into the curriculum, and finally we have $\mathcal{D}_o = [\mathcal{D}_{l1}, \cdots, \mathcal{D}_{lg}, \mathcal{D}_{r1}, \cdots]$. For notation convince, in the following, we use $\mathcal{D}_o = [\mathcal{D}_1, \cdots, \mathcal{D}_B]$ to denote the generated curriculum where $B = b + m$ denote the total number of datasets (opponent strategies).

**Loss Functions.** Before delving into the detailed training process, we first present the loss functions for training the Transformer model $M_\theta$ parameterized with $\theta$. For the AD algorithm, for each dataset $\mathcal{D}_i \in \mathcal{D}_o$, it includes $K$ rounds and each round $k$ has $T_k$ steps, i.e, $\left(s_0^{(k)} \right. =$

$(I_0^{(k)}, a_0^{(k)}, r_0^{(k)}), \cdots, s_{T_k}^{(k)} = (I_{T_k}^{(k)}, a_{T_k}^{(k)}, r_{T_k}^{(k)}))$. These data are concatenated into one *long* history, i.e., $\mathcal{D}_i = (s_0^{(1)}, \cdots, s_{T_1}^{(1)}, \cdots, s_0^{(K)} \cdots, s_{T_k}^{(K)})$. For convenience, we renumber this dataset, $\mathcal{D}_i = (s_0^{(i)}, \cdots, s_T^{(i)})$, where $T = \sum_{k=0}^{K} T_k$. Then the loss function for training the model $M_\theta$ with context length $L$ is given as

$$\mathcal{L}^{\text{AD}}(\theta) = -\mathbb{E}_{h_{t,L} \sim \mathcal{D}_i} \sum_{l=0}^{L} \log M_\theta(a_{t+l}^{(j)} | I_{t+l}^{(j)}, h_{t,l}^{(j)}) \tag{1}$$

where $h_{t,l} = (s_t, ..., s_{t+l})$ is the context data. Similarly, in the DPT algorithm, for each dataset $\mathcal{D}_i \in \mathcal{D}_o$, we have $\mathcal{D}_i = \{D_1 = (s_0^{(i)}, \cdots, s_T^{(i)}), D_2 = \{(I_j^q, a_j^*)\}_{j=0}^{K}\}$, where $D_1$ is the same as dataset with AD and $(I_j^q, a_j^*)$ is the query information set and its corresponding optimal action.

$$\mathcal{L}^{\text{DPT}}(\theta) = -\mathbb{E}_{(h_{t,L}, I^q, a^*) \sim \mathcal{D}_i} \sum_{l=0}^{L} \log M_\theta(a^* | I^q, h_{t,l}^{(j)}). \tag{2}$$

**Curriculum Training Process.** Given the generated curriculum $\mathcal{D}_o$, we sequentially train the Transformer $M_\theta$ on the datasets using the loss functions defined before, which is depicted in Algorithm 2. In this training process, a critical problem is catastrophic forgetting which is one of the common issues when learning on multiple tasks represented by the different datasets. To mitigate this issue, we propose to periodically retrain the model on the previously visited datasets. Intuitively, this review mechanism could ensure that the model retains its proficiency in the earlier learned tasks while simultaneously acquiring new capabilities for the new tasks. Let $\bar{\mathcal{D}}$ denote the set of datasets that have been visited. When $|\bar{\mathcal{D}}| < N$, we apply the review mechanism to determine the dataset in the current iteration $t$. With a slight abuse notation, we use $\mathcal{D}_t$ to denote the first dataset that has not been visited in $\mathcal{D}_o$. Then, at the current iteration $t$, we randomly sample a dataset from $\bar{\mathcal{D}}$ with probability $1 - \sigma$ while training on the current dataset $\mathcal{D}_t$ with probability $\sigma$. After all the datasets have been visited, i.e., $|\bar{\mathcal{D}}| = N$, we randomly sample a dataset from $\mathcal{D}_o$ at each training iteration.

---

**Algorithm 1** Curriculum Generation

1: **Input:** Learning-based generation $\mathcal{D}_l = [\mathcal{D}_{l1}, \cdots, \mathcal{D}_{ln}]$, random generation $\mathcal{D}_r = [\mathcal{D}_{r1}, \cdots, \mathcal{D}_{rm}]$, $g$, $\mathcal{D}_o \leftarrow \emptyset$
2: **for** $i = 1$ to $N$ **do**
3:     **if** $i \mod g = 0$ **then**
4:        $\mathcal{D}_o \leftarrow \mathcal{D}_o \cup \mathcal{D}_r[0]$, $\mathcal{D}_r \leftarrow \mathcal{D}_r \setminus \mathcal{D}_r[0]$;
5:     **else**
6:        $\mathcal{D}_o \leftarrow \mathcal{D}_o \cup \mathcal{D}_l[0]$, $\mathcal{D}_l \leftarrow \mathcal{D}_l \setminus \mathcal{D}_l[0]$;
7:     **end if**
8:     **if** $|\mathcal{D}_r| = 0$ or $|\mathcal{D}_l| = 0$ **then**
9:        $\mathcal{D}_o \leftarrow \mathcal{D}_o \cup \mathcal{D}_l$ or $\mathcal{D}_o \leftarrow \mathcal{D}_o \cup \mathcal{D}_r$;
10:      Early Break;
11:    **end if**
12: **end for**
13: **Output:** The curriculum $\mathcal{D}_o$

**Algorithm 2** Curriculum Training Framework

1: **Input:** Datasets $\mathcal{D}_o = [\mathcal{D}_1, ..., \mathcal{D}_N]$
2: Initialize Transformer $M_\theta$, $\sigma$ and $\bar{\mathcal{D}} \leftarrow \emptyset$;
3: **for** iteration $t = 1$ to $T$ **do**
4:     **for** train episode $p = 1$ to $M$ **do**
5:        **if** $|\bar{\mathcal{D}}| < N$ **then**
6:           Sample $\mathcal{D}_t$ from $\bar{\mathcal{D}}$ with prob. $1 - \sigma$ or use $\mathcal{D}_t$ in $\mathcal{D}_o$ with prob. $\sigma$;
7:        **else**
8:           Sample a dataset $\mathcal{D}_t$ from $\bar{\mathcal{D}}$;
9:        **end if**
10:      Train $M_\theta$ on $\mathcal{D}_t$ using Eq. (1) or (2);
11:    **end for**
12:    **if** $|\bar{\mathcal{D}}| < N$ **then** $\bar{\mathcal{D}} \leftarrow \bar{\mathcal{D}} \cup \{\mathcal{D}_t\}$; **end if**;
13: **end for**
14: **Output:** The Transformer $M_\theta$

---

### 4.3 STAGE III: INFERENCE

After the training process, we freeze the Transformer model and then utilize it for exploiting opponents as any player and computing the NE in equilibrium finding algorithms.

**Opponent Exploitation.** Given $M_\theta$, a direct application is to use the model to play as any player $p \in N$ and exploit any unknown opponent in an online manner over $K$ episodes of interaction. Let $C_p$ denote the context of player $p$, initialized to the empty set $C_p \leftarrow \emptyset$. For simplicity, let $E^j$ denote the complete trajectory of $j$-th episode. Then, in episode $k$, at each time step $t$, the context consists of the trajectories of the previous episodes and the interaction histories of the current episode $C_p = (E^1, \cdots, E^{k-1}, I_0, a_0, r_0, \cdots, r_{t-1})$. Next, an action is sampled according to $a_t \sim M_\theta(\cdot | I_t, C_p)$ for the current information set $I_t$. Finally, the interaction tuple $(I_t, a_t, r_t)$ is added to the context $C_p$ for predicting the next action in the next information set. After the $K$ episodes, we can use the final context $C_p$ to denote the "learned" strategy against the given opponent via in-context learning. For brevity, we use $C_p \leftarrow \text{OPPEXP}(M_\theta, C_{-p}, K)$ to denote the process of opponent exploitation against the opponent with strategy $C_{-p}$ and return $C_p$ the player $p$'s strategy.

**Equilibrium Finding.** We consider two common equilibrium-finding algorithms: self-play and PSRO. In the self-play framework, the model $M_\theta$ is used to play as each player in the game. Self-play takes turns to update the strategy of each player $p \in N$ through the above opponent exploitation process $C_p \leftarrow \text{OPPEXP}(M_\theta, C_{-p}, K)$ given the strategies of other players $C_{-p}$. However, the self-play framework does not guarantee convergence. Instead, a more popular framework is PSRO. We present the new PSRO framework in Algorithm 3, in which there are two primary differences compared to the conventional PSRO: i) As the model $M_\theta$ is frozen, the policy of each player $p \in N$ is represented by the context $C_p$, and thus, the (restricted) policy space of the player $p$ is a set of contexts $\Pi_p$; ii) Learning the best response policy of each player at each PSRO iteration is an opponent exploitation process given above.

---

**Algorithm 3** PSRO with Opponent Exploitation

---

1: **Input:** Transformer model $M_\theta$
2: Initialize players' policy spaces $\Pi_p = \{C_p \leftarrow \emptyset\}, \forall p \in N$;
3: **repeat**
4:     Update the meta-game payoff matrix $U^\Pi$ based on policy spaces $\Pi$ via simulation;
5:     Compute players' meta-strategies $\delta_p$ for $U^\Pi$ using any meta-solver, $\forall p \in N$;
6:     Expand policy space: $\Pi_p \leftarrow \Pi_p \cup \{\text{OPPEXP}(M_\theta, C_{-p}, K)\}$ where $C_{-p} \sim \delta_{-p}, \forall p \in N$;
7: **until** convergence
8: **Output:** The context sets and the meta-strategies of all players, $\Pi_p$ and $\delta_p, \forall p \in N$

---

## 5 EXPERIMENTS

To assess the effectiveness of **ICE** algorithm, we conduct comprehensive experiments on several popular extensive-form games. We start by outlining our experimental setting, followed by a detailed analysis of the results, structured around answering several key research questions.

### 5.1 EXPERIMENTAL SETTING

**Experimental Setup.** We selected a variety of poker games as test subjects, including both two-player and three-player versions of Kuhn Poker, Leduc Poker, and Goofspiel (with five cards). These games serve as diverse platforms to evaluate our algorithm's performance. i) To rigorously evaluate the performance of **ICE** algorithm in exploiting unknown opponents, we constructed three distinct types of testbeds by randomly sampling different opponents: *in-distribution*, *out-of-distribution*, and *NE opponent*. For the in-distribution testbed, we selected approximately 30 opponent tasks from the task dataset utilized during training. For the out-of-distribution testbed, we randomly sampled 20 opponent strategies to create a diverse set of test tasks. Finally, for the NE opponent testbed, we specifically configured the opponent's strategy to align with the NE strategy, thereby forming test tasks that directly reflect NE opponents. ii) To assess the performance of **ICE** algorithm in computing the NE strategy, we employ NASHCONV to measure the gap between the learned strategy and the NE strategy. Here, we focus specifically on two-player games as testbeds, as finding NEs in multi-player games is notably challenging.

**Baselines.** We conduct online testing against different opponents to evaluate the performance of **ICE** algorithm in exploiting unknown opponents. In this setting, we first select two widely used strategies as baselines: the Best Response (BR) strategy and the Nash Equilibrium (NE) strategy. Notably, BR is theoretically optimal in an online setting, as it is tailored to exploit a known opponent's strategy, making it a theoretical upper-bound benchmark. The NE strategy is included as a baseline due to its robustness against any opponent. Additionally, since this is an online setting, we select the proximal policy optimization (PPO) algorithm (Schulman et al., 2017) as another baseline for comparison. Furthermore, we include a multi-task pre-training with the fine-tuning framework as a baseline, as exploiting different opponents can be framed as a multi-task learning problem. Since BR and NE strategies are fixed once the opponent is given, we directly simulate these strategies against the opponent's strategy to evaluate their performance. For the PPO algorithm, multi-task pre-training with fine-tuning, and **ICE** algorithm, we conduct evaluations under a limited number of online interactions with the opponent. This limitation is deliberate, as our goal is to assess the capability of these algorithms to quickly and effectively exploit different unknown opponents.

## 5.2 EXPERIMENTAL RESULTS

To better illustrate our findings, we show them by answering the following research questions (RQs).

**RQ1:** *Can ICE act as any player in the game?*

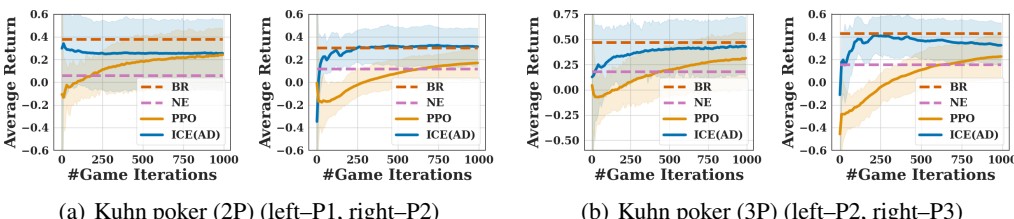

(a) Kuhn poker (2P) (left–P1, right–P2)    (b) Kuhn poker (3P) (left–P2, right–P3)

Figure 2: In-distribution results when acting as any player

We claim that **ICE** algorithm is capable of training a model to perform as any player in the game. To substantiate this claim, we conduct evaluations by positioning the model trained by the **ICE** algorithm in various player roles within the game. Fig. 2 presents the results for both two-player and three-player Kuhn poker, assessed using the in-distribution testbed. Our experimental results reveal that **ICE** outperforms both the NE strategy and the PPO algorithm when acting as any player of the game, whether in two-player or three-player games. Notably, **ICE** exhibits the capacity to self-improve and closely approximate the BR strategy, leveraging its in-context learning ability. These results show the effectiveness of our method in adjusting to various strategic roles. Consequently, we only show the results from the perspective of one player, as a representation of our algorithm's ability to adapt to and perform in any given role.

**RQ2:** *Can ICE adaptively exploit any opponent?*

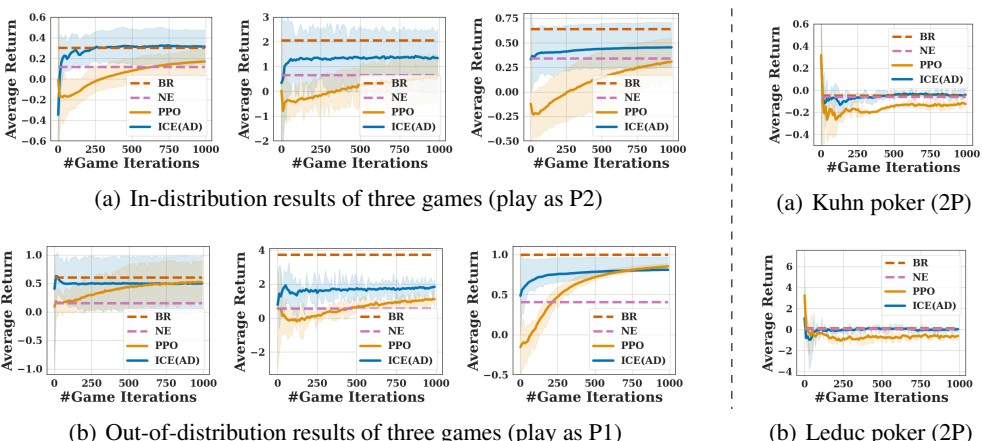

(a) In-distribution results of three games (play as P2)    (a) Kuhn poker (2P)

(b) Out-of-distribution results of three games (play as P1)    (b) Leduc poker (2P)

Figure 3: Results (left-Kuhn, middle-Leduc, right-Goofspiel)    Figure 4: NE opponent results

To answer this question, we perform experiments on the three distinct testbeds we previously introduced, which simulate different opponents including NE opponents. This diverse range of testing environments is crucial to comprehensively evaluate the adaptability and effectiveness of **ICE** algorithm in confronting any type of opponent. Fig. 3 and Fig. 4 display the results of playing three two-player games against different opponents. We have also carried out experiments for three-player games, with those results included in the Appendix due to page constraints. From Fig. 3, it is evident that **ICE** algorithm effectively demonstrates its in-context learning capability. Within a limited number of interactions, **ICE** surpasses both the NE strategy and the PPO algorithm. In simpler cases, the PPO algorithm may reach performance levels similar to that of **ICE** algorithm. A key distinction, however, is that unlike PPO and other RL algorithms which require retraining from scratch for each new opponent, **ICE** algorithm achieves this without any parameter updates. It is worth noting that the out-of-distribution results are worse than the in-distribution results, highlighting a key area for further improvement in enhancing the model's generalization capabilities. In Fig. 4, we observe

that **ICE** algorithm is capable of achieving results comparable to those of the NE and BR strategies, which means that **ICE** can achieve high rewards against the NE opponent.

**RQ3:** *Can **ICE** be used to compute NE without changing the parameter?*

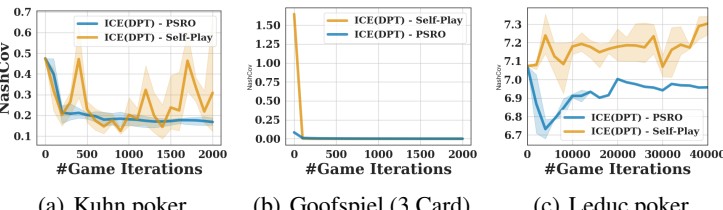

(a) Kuhn poker      (b) Goofspiel (3 Card)      (c) Leduc poker

Figure 5: Results on computing NE.

To answer this question, we conduct experiments on two-player Kuhn, Leduc, and Goofspiel poker games. We use the DPT algorithm to train the Transformer model, as its inference process does not impose sequence requirements on the context, making it convenient for evaluating the gap between the resulting strategy and the NE strategy, i.e., NASHCOV. Fig. 5 presents results using both the self-play and PSRO frameworks. From these results, we observe that, compared to the performance in Kuhn Poker and Leduc Poker, only the results in the Goofspiel game achieve low exploitability, regardless of whether the self-play or PSRO framework is used. This may be due to the existence of a pure strategy equilibrium in Goofspiel with 3 cards. Additionally, we observe that using the self-play framework to compute the NE strategy leads to unstable performance in Kuhn and Leduc Poker games. When using the PSRO framework, the NASHCONV decreases progressively in the Kuhn Poker game, indicating improved convergence, whereas it remains unstable in the Leduc Poker game. It may be the high dynamic in the Leduc Poker game. These findings represent a promising first step toward leveraging in-context learning for equilibrium computation without parameter updates. Nevertheless, the approach still struggles in games with complex dynamics, highlighting a clear direction for future work to improve robustness and adaptability in such scenarios.

**RQ4:** *How does **ICE** perform compared with multi-task pre-training with fine-tuning framework?*

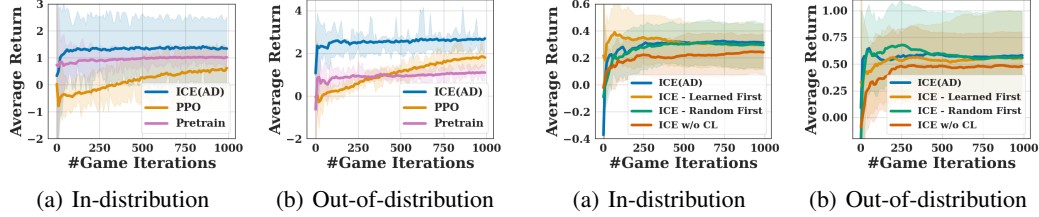

(a) In-distribution    (b) Out-of-distribution       (a) In-distribution    (b) Out-of-distribution

Figure 6: Leduc poker (2P) (play as P2)      Figure 7: Kuhn poker (2P) (play as P2)

Recent work has shown that multi-task pre-training with fine-tuning on new tasks performs equally or better than meta-learning pre-training with meta adaptation in RL tasks (Mandi et al., 2022). It indicates that pre-training with fine-tuning can quickly adapt to new tasks. In this paper, we compare this framework with **ICE** algorithm. Firstly, we pre-train a model using tasks generated from opponents' strategies, the same as those used in the **ICE** algorithm. Then, we evaluate its performance by fine-tuning based on the interactions with the opponent. The results for a two-player Leduc poker game are depicted in Fig. 6. Our findings reveal that **ICE** outperforms pre-training with fine-tuning approach in both in-distribution and out-of-distribution testbeds. Notably, pre-training with fine-tuning performs even underperforms compared to the PPO algorithm in the out-of-distribution testbed. It might be attributed to the extensive potential opponent strategies, where pre-training cannot encompass all opponent types, leading to slower adaptation to new tasks. Additionally, the conflict in training direction for different player roles in zero-sum games could further hinder the effectiveness of pre-training with fine-tuning.

**RQ5:** *Can our curriculum learning (CL) framework enhance the performance?*

**ICE** algorithm incorporates a curriculum learning (CL) framework for training the Transformer model. To explore the significance of CL, we conducted a comparative analysis by training the

Transformer model under different setups, including without the CL framework, which is trained on a randomly ordered sequence of tasks, training first on random opponents, followed by learning-based opponents ("Random First"), and training first on learning-based opponents, followed by random opponents ("Learned First"). The results are presented in Fig. 7. Interestingly, **ICE**, Random First, and Learned First achieve similar overall performance, which may be attributed to the inherent curriculum-like nature of learning-based opponents. However, we observe that Learned First performs better in in-distribution cases, likely due to the stronger initial focus on structured strategies and Random First performs better in out-of-distribution cases, benefiting from early exposure to diverse random strategies. Our **ICE** method demonstrates more stable performance across both in-distribution and out-of-distribution settings. This differential in performance underscores the significant contribution of the CL framework in boosting the effectiveness of **ICE** algorithm.

**RQ6:** *Does the context length influence the performance of ICE?*

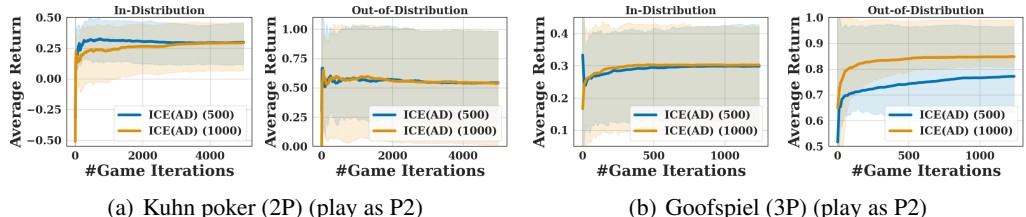

(a) Kuhn poker (2P) (play as P2)  (b) Goofspiel (3P) (play as P2)

Figure 8: Results of different context lengths

To investigate how the pre-defined context length affects performance, we conducted experiments with various context lengths in our game scenarios. The results for two-player Kuhn poker and three-player Goofspiel are shown in Fig. 8. For Kuhn poker, we observe that context length has minimal impact on performance. This could be attributed to the simplicity of the game, where even a short context is sufficient for effective in-context learning. Additionally, we note that, in the early stages, a larger context length may initially underperform compared to a shorter one, possibly because more interactions are required to fully leverage the extended context. Conversely, for the Goofspiel game, the results indicate that a larger context length improves performance. It implies that in more complex games, a large context, which includes more interaction history information, can significantly enhance the decision-making process. The extended context length provides a broader historical perspective, which is particularly beneficial in complex strategic environments where previous interactions greatly influence future decisions.

## 6 CONCLUSION

In this paper, we investigate an important research question: *Can we leverage ICL to learn a model to i) play as **any player** of the game, ii) exploit **any opponent** to maximize the utility, and iii) be used to compute NE, **without changing the parameters**?* To this end, we propose In-Context Exploiter (**ICE**), which aims to train a single model that can satisfy all the desiderata presented in the previous question. **ICE** delivers three main contributions: i) it generates diverse opponents via the equilibrium finding algorithms, e.g., CFR and PSRO, and collects the trajectories of players using PPO as the training dataset; ii) it combines the curriculum learning and ICL for single-agent scenarios (AD and DPT) to train the model and employs a revisiting mechanism to preventing catastrophic forgetting when training the model on multiple datasets; iii) it leverages the trained model to play as each player in opponent exploitation and integrates the trained model into equilibrium finding algorithms, e.g., PSRO, to compute NE of the game. Extensive experimental results demonstrate that **ICE** can efficiently exploit different opponents and can be used to compute NE without updating parameters.

**Limitations and Future Work.** There are several limitations to this work. First, we only focus on the games which are relatively small-scale. The ICL for large-scale games would require large models with longer in-context lengths and training time. Second, for the equilibrium finding with the trained model, we only focus on NE. Other solution concepts such as (coarse) correlated equilibrium (CCE) will be considered for multiplayer games. Third, the theoretical analysis of the ICL for games is largely unexplored, which will be investigated in future work. More detailed discussions on the limitations and future directions can be found in Appendix A.4.

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

# A   DISCUSSION

## A.1   **ICE** VS. OPPONENT MODELING

ICE can be viewed as a method with **implicit** opponent modeling (He et al., 2016; Albrecht & Stone, 2018), where the modeling of the opponent is implicitly encoded into the parameters of the model. There are several advantages of **ICE** over opponent modeling: i) **ICE** does not need an explicit model for the opponents, where the explicit model in the opponent modeling may restrict the generalizability of the methods, ii) **ICE** can exploit different opponents without changing the parameters, where the opponent modeling may need to fit the parameters of the opponent model during game play and then make the decision in response to the opponent. To summarize, **ICE** is simpler, more efficient, and more generalizable. **ICE** also has the disadvantage, i.e., the ability to model the opponents is largely determined by the length of the in-context. With longer in-context, **ICE** can model more opponents, while the model will also be larger and the training cost will be increased. We will discuss the methods to reduce the length of the in-context in the next section. On the other hand, we can also introduce an explicit model for opponents into ICE, where the parameters of the opponent model can be fitted through in-context learning. The explicit opponent model can help us to understand the internal mechanism of ICE.

## A.2   **ICE** VS. EQUILIBRIUM FINDING

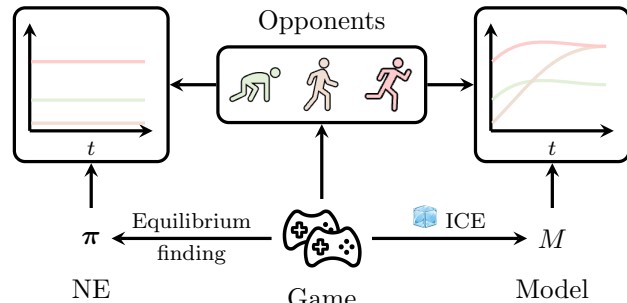

Figure 9: Comparison between equilibrium finding and our method

The **ICE** approach and traditional equilibrium finding share the goal of developing strategies for extensive-form games but diverge significantly in their methodology and capabilities. Equilibrium finding methods focus on computing an NE strategy profile $\pi$ that performs robustly against any potential opponent. This is illustrated in Fig. 9 (left), where the NE strategy remains stable over time, providing consistent performance across a wide range of opponents. In contrast, the **ICE** framework is designed to exploit any opponent effectively by adapting its strategy through in-context learning. As shown in Fig. 9 (right), **ICE** leverages the trained model $M$ to exploit different opponents only through in-context learning, leading to performance that improves dynamically over time as more interactions are observed. This adaptability allows **ICE** to achieve better performance against other irrational opponents compared to the fixed NE strategy.

## A.3   **ICE** VS. ONLINE LEARNING, MULTITASK LEARNING, AND META LEARNING

ICE, as well as other in-context learning methods (Laskin et al., 2022; Lee et al., 2024), is similar to online learning methods, e.g., no-regret learning (Shalev-Shwartz et al., 2012). However, **ICE** does not change the parameters of the model during the game play with the opponents, which differs from online learning. We believe that online learning, especially no-regret learning, can be used to analyze the behaviors of **ICE** and in-context learning methods, which will be explored in future works. We also consider online learning methods as our baselines. We note that PPO is an online learning and on-policy method and PPO is scalable and widely used. Therefore, we include PPO as the baseline in our experiments.

The training of **ICE** is also similar to multitask learning (Mandi et al., 2022) and meta-learning (Finn et al., 2017), where multi-task learning learns a policy for different tasks, and meta-learning enables

the fast adaption of the learned policy on specific tasks. **ICE** also learns a policy for different tasks, where the model parameters are not changed, but the behaviors are changed during game play. In the experiment section, we choose the PPO method initialized with a pre-trained policy to benchmark multi-task and meta-learning methods, as shown in Figure 6.

### A.4  LIMITATIONS AND FUTURE WORKS

In this section, we provide a mode detailed discussion about the limitations of this work.

**Scalability of ICE.** We only focus on relatively small-scale games in this work, including Kuhn poker and Leduc poker. However, the large-scale games will have more relevance for real-world deployment, such as Texas Hold'em poker (Brown & Sandholm, 2019) and sport games, e.g., football game (Wang et al., 2024). We will consider to scale up the current **ICE** methods to tackle large-scale games, which may require larger models with longer in-context lengths and longer training time. We would also consider to train the foundation model of ICL for games, i.e., one model trained to generalize to different games to further improve the generalizability of ICL.

**Other Solution Concepts.** We only focus on exploiting the opponents and computing NE in this work. However, there are many other solution concepts such as (coarse) correlated equilibrium (CCE) and quantal response equilibrium (QRE) (McKelvey & Palfrey, 1995), which are also important concepts in game theory. **ICE** has the potential to compute other solution concepts with one single pre-trained model, which will be tacked in future work.

**Theoretical Analysis.** The ICL for games is a new research area and the theoretical analysis is required for further investigations, including the optimality of the converged solutions and the convergence of the equilibrium finding with the ICL models. In the future, we will conduct a systematic theoretical analysis of the ICL for games, similar to the analysis for DPT (Lee et al., 2024).

## B    IMPLEMENTATION DETAILS

In this section, we provide the experimental details of **ICE** algorithm from its three main stages.

**Opponent Generation.** In this paper, we employ two methods, as introduced in the main paper, to generate a diverse range of opponent strategies. To implement the random generation method, we traverse through all the information sets of an opponent and assign a randomly generated strategy to each information set. This approach allows us to generate various opponents exhibiting random behaviors. To implement the learning-based generation method, we utilize a well-known algorithm, Counterfactual Regret Minimization (CFR) (Zinkevich et al., 2007), as the equilibrium-finding algorithm. By applying CFR to solve the game, we record the average strategy for each player at each iteration. This process generates a series of opponent strategies that evolve from random to increasingly robust over time. These two methods collectively ensure that our dataset includes a wide spectrum of opponent strategies, ranging from entirely unpredictable to highly strategic. Such a comprehensive dataset is instrumental in training our model to adapt and respond effectively to various levels of opponent sophistication and strategy.

**Interactive History Collection.** It's important to recognize that when an opponent's strategy is known, the task of exploiting that opponent to maximize utility effectively becomes a reinforcement learning (RL) problem. Consequently, each distinct opponent strategy corresponds to a unique RL task. For the AD algorithm, to collect interactive history data from our diverse opponent strategies for training purposes, we adopt the Proximal Policy Optimization (PPO) algorithm (Schulman et al., 2017) to address each of these RL tasks. During this process, we systematically record the learning history of the PPO algorithm, specifically capturing the contents of the reply buffer used by PPO.

For the DPT algorithm, the training dataset consists of three components: in-context data (i.e., interactive history data), query states, and optimal actions. To generate the in-context data, we use a random strategy to play against various opponents and record the resulting interactive history. The optimal strategies used against different opponents are derived from the PPO policies that were trained earlier. Subsequently, we randomly sample query states and utilize the optimal strategy to obtain the corresponding optimal actions.

**Curriclum Learning.** The curriculum learning framework is crucial for effectively training the Transformer model, with the curriculum's design being its core component. While the main paper provides a comprehensive explanation of the curriculum generation process and the overall learning framework structure, this section will not revisit those specifics. However, it is important to highlight that the thoughtful design of the curriculum is key to the success of the model's training. By gradually increasing task complexity and progressing through structured stages, the model is able to incrementally build its understanding and capabilities without being overwhelmed. This approach aligns well with the principles of in-context learning, allowing the Transformer to adapt and respond efficiently to a broad spectrum of strategic scenarios.

**Inference.** In the main paper, we introduce how our trained model is used to exploit opponents and compute equilibrium strategy. Here, we provide a detailed description of the inference process. We first focus on how to use our model to exploit opponents, as outlined in Algorithm 4. For a given opponent, the model $M_\theta$ is inferred based on the current context to make decisions, while simultaneously updating the context with the previous interactive history data.

Next, we introduce two frameworks for computing equilibrium strategies, detailed. The first framework is self-play, where the model essentially plays against itself by taking on the roles of all players in the game. As described in Algorithm 5, the model plays each player in turn to exploit the opposing strategies. It is important to note that the context $C$ is sufficient for recording strategies, as the strategy for any state $s$ can be queried through $M_\theta(\cdot|s, C)$. Therefore, only the most recent context needs to be stored. The second framework is PSRO, a widely-used algorithm for solving imperfect-information extensive-form games. In this framework, we substitute the best response oracle with the opponent-exploiting inference process. The context is recorded to represent the learned best response strategy effectively.

**Parameter Setting.** Here, we list the parameters used in the **ICE** algorithm for all games in Tab.2. In this table, the previous rate $\sigma$ is used to control the blend of new and prior tasks to prevent catastrophic forgetting and the number of trains per task refers to the number of training for each selected task (i.e., $M$ in Algorithm 2).

---

**Algorithm 4** Opponent Exploiting (OPPEXP($M_\theta, C_{-p}, K$))

---

1: **Input:** Transformer model $M_\theta$, player id $p$
2: Initialize the context $C = \emptyset$;
3: **for** $t = 1$ to $K$ **do**
4:    $s$ is initialized as game's initial state and $s_{pre}, a_{pre} =$None, None;
5:    **while** TRUE **do**
6:      **if** $s$ is the $P_i$'s turn **then**
7:        **if** $s_{pre}$ is not None **then**
8:          ADD $(s_{pre}, a_{pre}, r(s), d(s))$ to $C$;
9:        **end if**
10:       $a = M_\theta(\cdot|C, s)$;
11:       $s_{pre} = s, a_{pre} = a$;
12:      **else**
13:       # get the opponent action
14:       $a = M_\theta(\cdot|C_{-p}, s)$;
15:      **end if**
16:      # get the next game state based on transition function
17:      $s' = T(s, a)$;
18:      **if** $s'$ is the end state **then**
19:       ADD $(s_{pre}, a_{pre}, r(s), d(s))$ to $C$;
20:       Break;
21:      **end if**
22:    **end while**
23: **end for**
24: **Output:** Context $C$

---

**Algorithm 5** Self-Play Framework

---

1: **Input:** Transformer model $M_\theta$, Iteration number $K$
2: **for** $i = 1$ to $T$ **do**
3:    **for** $p$ in $\{1, ..., n\}$ **do**
4:      $C_p =$ OPPEXP($M_\theta, C_{-p}, K$);
5:    **end for**
6: **end for**

---

Table 2: Parameter for ICE(AD)

| Games | Kuhn | Leduc | Goofspiel | Kuhn | Leduc | Goofspiel |
|---|---|---|---|---|---|---|
| Number of Player | 2 | 2 | 2 | 3 | 3 | 3 |
| Previous Rate $\sigma$ | 0.1 | 0.3 | 0.3 | 0.1 | 0.3 | 0.3 |
| Number of Train per Task $M$ | 10 | 10 | 10 | 10 | 30 | 30 |
| Context Length | 1000 | 1000 | 1000 | 1000 | 1000 | 1000 |

## C ADDITIONAL EXPERIMENTAL RESULTS.

In this section, we present further experimental results to substantiate the effectiveness of **ICE** algorithm in opponent exploiting. While the main paper provided the performance of **ICE** in three two-player game scenarios evaluated across three distinct testbeds, here we extend our analysis to include results from experiments conducted on three different three-player games.

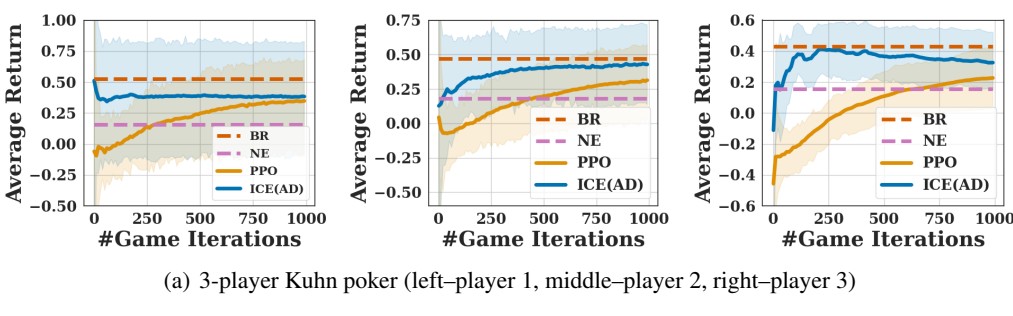

(a) 3-player Kuhn poker (left–player 1, middle–player 2, right–player 3)

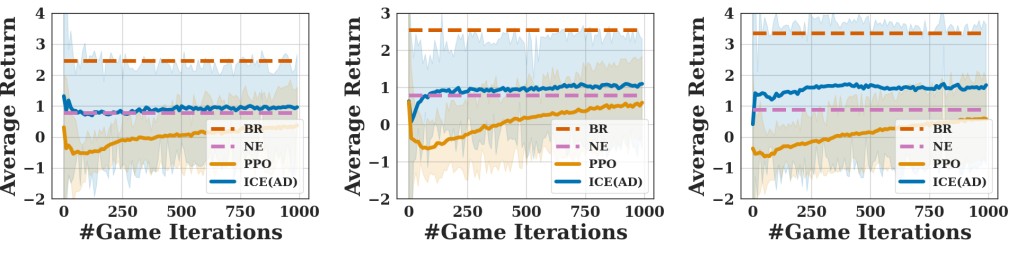

(b) 3-player Leduc poker (left–player 1, middle–player 2, right–player 3)

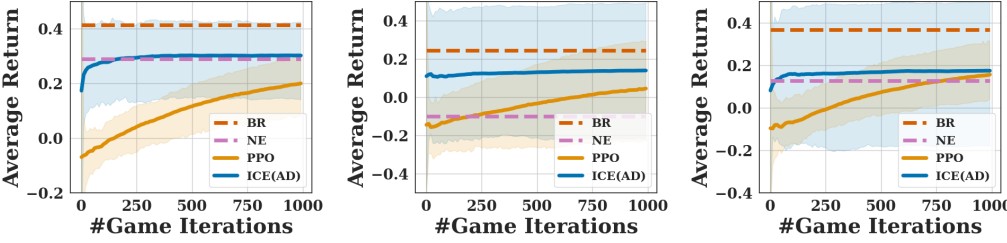

(c) 3-player Goofspiel (left–player 1, middle–player 2, right–player 3)

Figure 10: In-distribution results of three-player games

Firstly, we present the results from the in-distribution testbed in Fig.10. In these three three-player games, it is evident that the model trained using the **ICE** algorithm successfully functions as any player in the game, demonstrating in-context learning ability with increasing iterations. The Best Response (BR) strategy, while theoretically the optimal approach since it is tailored against a known opponent's strategy, isn't practical in real-world scenarios where an opponent's strategy isn't known in advance. In our results, the BR strategy's performance is included merely as a theoretical benchmark. Notably, while the ICE-trained model doesn't achieve the theoretical optimal values of the BR strategy, it consistently surpasses both the NE strategy and the PPO algorithm. This observation is significant as it indicates that the ICE-trained model can exploit the opponents more effectively than the NE strategy, which is generally considered the most conservative approach. The ability of **ICE** to outperform in these multi-player game scenarios demonstrates its potential as a powerful tool for strategic decision-making in complex, real-world situations.

Next, Fig.11 shows the results from the out-of-distribution testbed, where we observe trends similar to those in the in-distribution testbed. The key distinction here is that the opponents in the out-of-distribution testbed are randomly generated, which often results in simpler strategic scenarios. In contrast, the in-distribution testbed encompasses a mix of randomly generated and learning-

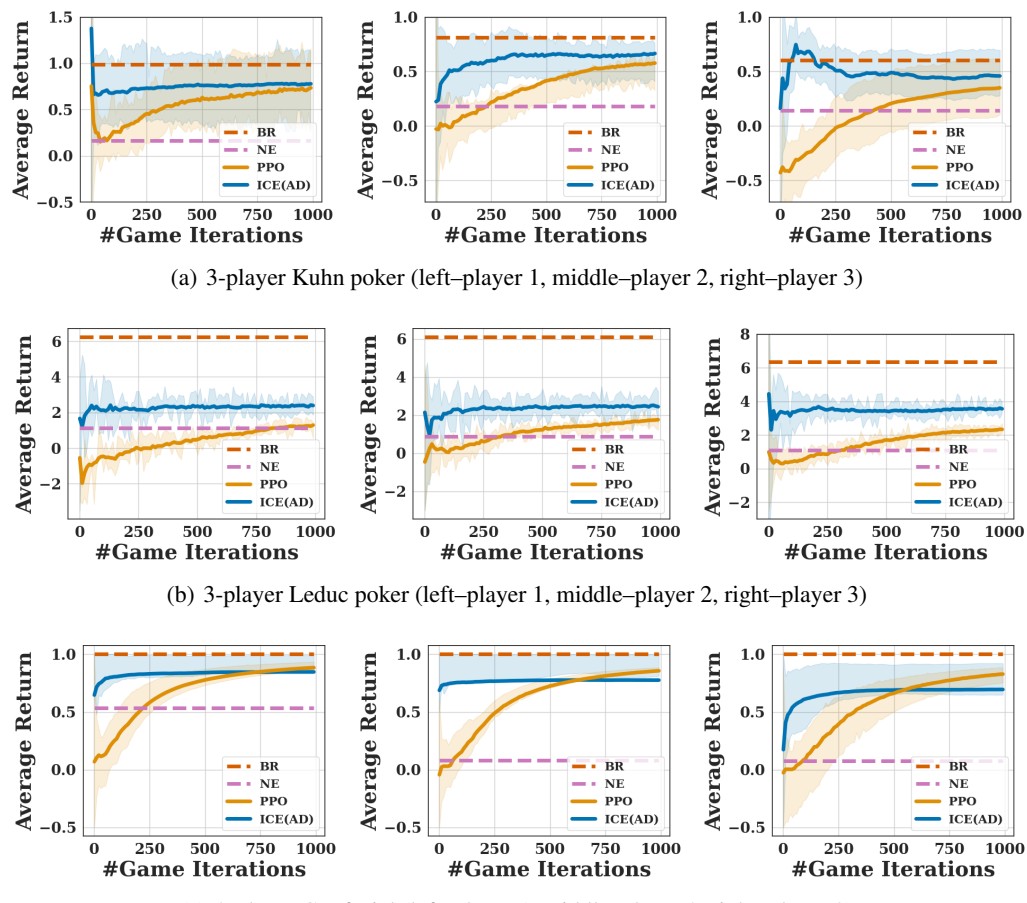

(a) 3-player Kuhn poker (left–player 1, middle–player 2, right–player 3)

(b) 3-player Leduc poker (left–player 1, middle–player 2, right–player 3)

(c) 3-player Goofspiel (left–player 1, middle–player 2, right–player 3)

Figure 11: Out-of-distribution results of three-player games

generated opponents, leading to potentially more complex and challenging interactions. An interesting observation in the three-player Goofspiel game is that, after 500 interactions, the PPO algorithm begins to match the performance of ICE. This trend could be attributed to the simpler nature of the randomly generated opponents in the out-of-distribution testbed, which might be easier for PPO to adapt to and exploit over time. Despite this, **ICE** demonstrates a faster convergence to high-performance levels compared to the PPO algorithm and consistently outperforms the NE strategy. It indicates that **ICE** is not only capable of quickly adapting to new opponents but also effectively maximizing performance in diverse opponent settings, including both simple and complex strategic environments.

Lastly, we discuss the results against NE opponents, as shown in Fig.12. Our findings reveal that the **ICE** algorithm achieves better or comparable performance to the NE strategy only in the three-player Kuhn poker game. However, in other game scenarios, while **ICE** does not outperform the NE strategy, it still maintains a higher level of performance than the PPO algorithm. The less optimal performance of **ICE** in these cases can be attributed to the highly dynamic game environment and stability of the NE opponents. In three-player games, the player faces two opponents simultaneously, and if both adopt the conservative NE strategy, exploiting them concurrently becomes significantly challenging. This observation highlights an area for future development. Improving the **ICE** algorithm to more effectively handle situations where multiple opponents employ highly conservative strategies, such as the NE, will be a focus of our future research.

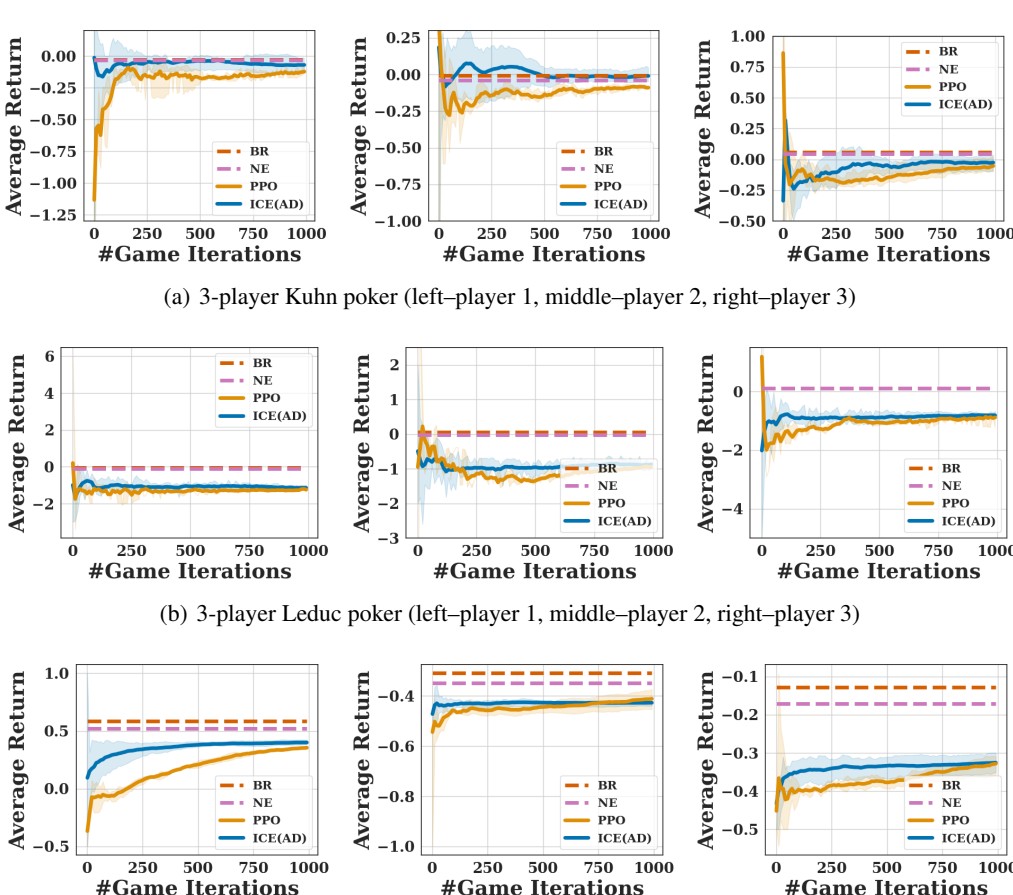

(a) 3-player Kuhn poker (left–player 1, middle–player 2, right–player 3)

(b) 3-player Leduc poker (left–player 1, middle–player 2, right–player 3)

(c) 3-player Goofspiel (left–player 1, middle–player 2, right–player 3)

Figure 12: Results of three-player games against NE opponent

