# OpenReview forum: "In-Context Learning for Games"
_ICLR.cc/2025/Conference — Submitted to ICLR 2025_

### Official Review · Reviewer_wMz9 · 2024-11-01

**Soundness:** 2
**Presentation:** 3
**Contribution:** 2
**Rating:** 3
**Confidence:** 4

**Summary:**

The authors introduce In-Context Exploiter (ICE), an algorithm addressing three questions: 1) Can in-context learning (ICL) enable a model to act as any player in a game? 2) Can it exploit opponents for maximum utility without updating parameters? 3) Can it compute the Nash Equilibrium (NE) with the same model? They use curriculum learning (CL) and reinforcement learning (RL) to train a highly adaptable Transformer model.

While this paper has limited novelty, rejection is still recommended for those listed in weaknesses.

**Strengths:**

1. Innovation: Combining NE and RL to generate diverse opponent data and proposing the ICE algorithm based on the ICL method, providing a complete framework from data collection and training to inference. But, it is common for ??? Therefore, the novelty is limited.

**Weaknesses:**

1. **Lack of Clarity in Algorithm 3 Evaluation**: It is unclear what specific advantages are demonstrated by PSRO with Opponent Exploitation in Algorithm 3. Is the primary goal to show that it can quickly approximate NE?
2. **Lack of Comparisons to Other Methods of Using ICL with RL**: There're a lot of existed ICL x RL methods, what's the main difference between yours and others? Given that ICL is applied to adapt to various opponent strategies, would directly integrating ICL into the RL training process yield better results? For instance, using historical **trajectories** to inform ICE in determining opponent strategies and providing this data to an RL model might enhance opponent exploitation.
3. **Unconvincing Experiments:** The experiments compare ICE (pretrained model) with PPO (Non-pretrained model), which is trivial that pretrained model can perform better at the previous time step. And in Figure 4, the performance drops afterwards. Results indicate that ICE performs slightly worse in OOD settings and is even surpassed by RL methods within 1000 iterations in some cases.
4. **Limited Ablation Study**: The study lacks ablation experiments on multiple factors, making it unclear whether ICE outperforms multi-task pre-training with fine-tuning frameworks in OOD  scenarios.  Moreover, the study does not clearly address the diversity or strength of opponent strategies, which raises questions about the method's potential for surpassing RL in broader applications.
5. **Lack of Training Details**: The paper does not provide sufficient information on the amount of data, the number of training iterations required, or the specific model configurations.

**Questions:**

1. **Terminology Consistency and Figure Check**: In the paper, terminology is inconsistent; for instance, Section 4.2 refers to the "Curriculum Training Process," Algorithm 2 labels it as "Curriculum Training Framework," while in Section 5.2, it appears as "curriculum learning (CL) framework." If these refer to the same concept, a unified term should be used; if not, a clear distinction should be provided. Additionally, Figure 9 may contain an error, as the text states that the model "when combined with CL framework outperforms the version without CL," but the figure seems to indicate that w/ CL reward is lower.
2. **Provide Training Details and Data Requirements**: According to the results, the paper aims to demonstrate ICE's rapid adaptability across various opponents in specific game scenarios, which seems to benefit from pre-designed diverse opponent strategies. In contrast, RL methods may not require such pre-training. It would be helpful to further compare ICE's training requirements with RL methods (e.g., multi-task pre-training or representative NE methods within equilibrium finding), to substantiate ICE's claims of reduced training and application costs relative to RL or PSRO methods.
3. **Enhanced Ablation Studies**: The experiments currently only compare specific variables, but there is a lack of a comprehensive comparison across multiple factors. It is recommended to include tables for variable comparison or add parameter settings in the main text (some of which are found in B IMPLEMENTATION DETAILS). Ablation studies should compare different context lengths, game types (including player positions), and results for in-distribution vs. out-of-distribution players, to further illustrate ICE's early-stage training advantages over RL methods.

---

> ### Author Response · Authors · 2024-11-25
> **Reply to Reviewer wMz9**
>
> Thank you for your valuable feedback. Below is our response to your question:
>
> ---
>
> **Weakness 1**
>
> - Algorithm 3 integrates opponent exploitation within the PSRO framework. In this design, opponent exploitation replaces the best response oracle in the vanilla PSRO algorithm, offering a more efficient alternative by reducing computational overhead.
>
>
> - The key advantage of Algorithm 3 is its ability to function without requiring parameter updates. In vanilla PSRO, the best response oracle involves training BR policies, which can be computationally expensive. In contrast, Algorithm 3 utilizes in-context learning to derive the BR policy directly, eliminating the need for parameter updates and significantly enhancing efficiency.
>
> ---
>
> **Weakness 2**
>
> - Indeed, integrating ICE into RL could enhance opponent exploitation. However, a traditional RL algorithm is typically designed to optimize the utility of a single player given a fixed opponent strategy. In this paper, our goal is to train a model capable of playing as any player in the game, enabling it to exploit its opponent while also contributing to equilibrium strategy computation.
>
> - If ICE were integrated into RL, the RL framework would be limited to optimizing for a single player’s perspective, reducing its ability to generalize across different player roles in the game. Therefore, this method cannot solve our problem.
>
> ---
>
>
> **Weakness 3**
>
> - We agree that comparing a pre-trained model (ICE) with a non-pre-trained model (PPO) might seem to give ICE an inherent advantage at earlier time steps. However, the purpose of this comparison is to demonstrate the ability of ICE to adapt efficiently to different opponent strategies using in-context learning without parameter updates, as opposed to PPO, which requires continuous parameter optimization.
>
> - The performance drop observed in Figure 4 and the slightly worse performance of ICE in out-of-distribution (OOD) settings are important findings. These results indicate that while ICE excels in in-distribution settings, its adaptability to OOD opponents and long-term performance can be further improved.
>
>
> ---
>
> **Weaknesses 4, 5, and Question 3**
>
> - In the paper, we have conducted ablation studies on different context lengths (RQ6) and different player positions (Appendix C). All these results are evaluated in both the in-distribution and out-of-distribution testbeds. Additionally, Figure 8 shows that ICE outperforms multi-task pre-training with fine-tuning frameworks in OOD scenarios in the Leduc poker game.
>
> - For parameter settings, we have provided the necessary parameter table in Appendix B. In future revisions, we will consider including key parameters directly in the main paper to improve accessibility and clarity.
>
> ---
>
> **Question 1**
>
> - Thanks for pointing out. These terms refer to the same concept. We have unified the terminology throughout the paper for clarity in the revision. In Figure 9, it should be “w/o” instead of “w/” and we have corrected it in the revision.
>
> ---
>
> **Question 2**
>
> - ICE benefits from pre-designed diverse opponent strategies to train a transformer-based model capable of in-context learning. This design provides several advantages, including rapid adaptability and parameter-free inference during deployment. Although these benefits come at the cost of requiring a curated training dataset, which may involve computational overhead during pre-training, ICE has more advantages than RL and equilibrium-finding algorithms.
>
> - RL approaches, such as PPO, optimize policies online and do not require pre-designed opponent datasets. However, they often involve continuous parameter updates and can incur significant computational costs during deployment, especially when adapting to new opponents. Contrast this with ICE, ICE shifts computational effort to the pre-training phase, enabling faster adaptation during inference.
>
> - PSRO typically requires iterative computation of best responses and meta-strategies, which can be computationally expensive, especially in large games or with complex opponent pools.	In contrast, ICE uses in-context learning to approximate best responses without requiring parameter updates, potentially reducing application costs.
>
> ---
> We hope our response can address your concerns, and we deeply appreciate that if you could reconsider the evaluation of our paper.

---

> > ### Comment · Reviewer_wMz9 · 2024-12-02
> >
> > Thank you for your response. I decide to maintain the current rating for the following reasons:
> >
> > 1.	Regarding your response to Weakness 1, the experimental design focuses on performance rather than efficiency. We suggest providing a detailed comparison in the paper to make it more convincing.
> >
> > 2.	As for your response to Weakness 2, where you claim that RL is not well-suited for solving the problem, we remain skeptical. Unless more decisive experimental evidence is provided, this claim is difficult to accept.
> >
> > 3.	In response to your explanations for Weakness 3 and Question 2, we still believe that the advantages of ICE are not particularly apparent.
> >
> > Thus, I would keep my rating (3).

---

> > > ### Author Response · Authors · 2024-12-02
> > >
> > > Thanks for your detailed feedback and for taking the time to review our responses.
> > >
> > > ---
> > > **Regarding Weakness 1**
> > >
> > > We respectfully argue that the efficiency of Algorithm 3 is demonstrated through its ability to adapt without requiring parameter updates.  This stands in contrast to traditional methods for computing the best response within the PSRO framework, such as RL, which necessitate frequent parameter updates when computing the best response strategy.  By eliminating the need for parameter updates, ICE significantly reduces computational overhead, highlighting a key aspect of efficiency.  We will include a more detailed discussion about this efficiency in the revised version of the paper.
> > >
> > > ---
> > > **Regarding Weakness 2**
> > >
> > > We would like to clarify that while integrating ICL with RL algorithms can effectively optimize from the perspective of a single player, our goal is to develop a model that can play as any player in the game, exploit opponents, and compute NE without parameter updates. Clearly, directly applying ICL+RL algorithms to games would not address this broader goal.
> > >
> > > Our work extends ICL+RL algorithms, including Algorithm Distillation (AD) and Decision Transformer (DPT), to the domain of multi-player games. This extension required several key contributions, including:
> > >
> > > - Designing methods to generate diverse opponent strategies, which allows us to frame opponent exploitation as an RL problem.
> > >
> > > - Proposing a framework that efficiently computes NE strategies without parameter updates by using the In-context ability of the model.
> > >
> > > By combining these contributions, our approach generalizes the application of ICL+RL methods to the multi-player game area, enabling both opponent exploitation and NE computation.
> > >
> > > ---
> > > **Advantages of ICE**
> > >
> > > We would like to emphasize that applying in-context learning (ICL) to games is still a relatively new and underexplored research area. Our work represents an initial step toward investigating how ICL can be effectively applied in this domain. One of the key advantages of ICL is its ability to adapt without parameter updates, which can address a major challenge in the game domain – specifically, the repeatedly computing BR during NE computation.
> > >
> > > In terms of ICE’s specific advantages:
> > >
> > > - ICE enables efficient adaptation to diverse opponents during inference, making it particularly suitable for scenarios requiring rapid response without additional training.
> > > - ICE integrates opponent exploitation and NE computation into a pre-trained model, offering practical adaptability and efficiency.
> > > - ICE operates without parameter updates, significantly reducing computational overhead compared to traditional methods that rely on frequent parameter updates for BR computation.
> > >
> > > While our results show promising progress, we acknowledge that there are areas for improvement, particularly in handling games with complex dynamics. These findings highlight the potential of ICE while pointing to future directions to further enhance its robustness and applicability.
> > >
> > > ---
> > >
> > > We hope these responses address your concerns and welcome any additional questions or feedback you may have.

---

> ### Author Response · Authors · 2024-12-02
> **Follow-Up on Review Discussion**
>
> Dear Reviewer wMz9,
>
> Thank you once again for your valuable review.
>
> As the author-reviewer discussion period is nearing its end, we would like to kindly ask if our responses have addressed your concerns. We have put a lot of effort into preparing detailed responses to your questions. We are eager to address any additional feedback or unresolved concerns you may have before the discussion period concludes.
>
> We look forward to hearing from you. Thank you for your time and consideration.
>
> Sincerely,
>
> Authors of Submission 8911

---

> ### Author Response · Authors · 2024-12-03
>
> Dear Reviewer wMz9,
>
> Thank you for your time, effort, and thoughtful feedback in reviewing our work. As the author-reviewer discussion period approaches its conclusion (11:59pm AoE on 2nd Dec), we sincerely hope that our responses have addressed your concerns.
>
> If our responses have adequately resolved your concerns, we would be deeply grateful if you could reconsider your evaluation of our submission. However, if there are any remaining issues or uncertainties that you feel we have not sufficiently addressed, we kindly ask for further clarification or suggestions. We are eager to make additional improvements before the discussion period ends to ensure our work meets your expectations.
>
> Thank you again for your valuable feedback and for helping us improve our paper.
>
> Sincerely,
>
> Authors of Submission 8911

---

### Official Review · Reviewer_XVLk · 2024-11-03

**Soundness:** 2
**Presentation:** 2
**Contribution:** 2
**Rating:** 3
**Confidence:** 4

**Summary:**

This paper proposes ICE, an in-context learning framework for multiagent games. Specifically the method consists of three stages, 1) collect interaction history from diverse opponents 2) optimise a transformer network to predict next action based on cross-episode interaction history with diverse opponents and 3) use the trained transformer policy network to act at inference time without further training.

The key property of ICE is that the transformer network is no longer trained at inference time and only game interaction context is needed.

**Strengths:**

Leveraging transformer architecture for in-context learning in (single agent-) games has been a popular direction and the authors here extend this direction to multiplayer games and provided comprehensive experiments substantiating their claims that ICE is performant, does not require per-opponent re-training and can generalise across diverse opponents.

**Weaknesses:**

I'm unfortunately quite confused by the motivation for using transformer's in-context learning capabilities for the games that have been studied and I also find some of the experiments difficult to follow.

On the former, the in-context learning capability of transformer is typically studied in the context of transformer models that have been pre-trained on general domain datasets (e.g. Internet text corpus). By virtue of the generality of the training data and the effectiveness of the transformer architecture, one could leveraged such pre-trained models for in-context learning in domains where common sense knowledge is traditionally difficult to come by (e.g. SayCan [1] in robotics). If I understood correctly, the authors do not leverage pre-trained general-domain knowledge but generate domain-specific data that are used to train a specialist transformer model. From that perspective, "in-context learning" should be compared to meta-learning algorithms such as $RL^2$ [2] where the policy is conditioned on cross-episode interaction history. In multiplayer games, prior work at ICML such as $simplex-NeuPL$ [3] also showed that a policy network can adapt to diverse opponents, achieve near optimal returns (compared to BR policies) without inference time training. Neither of these works utilised the transformer architecture but I wouldn't find it surprising the a transformer architecture, with carefully constructed training dataset, would be able to achieve similar results.

On the latter, the authors did comprehensively study several experimental settings but I find the description of the experiments difficult to follow. For example, the authors compared ICE (AD) with PPO and found that PPO could not achieve optimal returns (compared to BR). However, it isn't clear what "assessed using the in-distribution testbed" would mean in this context. Is it a single opponent policy that's being played by the opponent? Or is it a distribution over opponent that's being sampled? For ICE (AD), are the "in-contexts" generated by one opponent policy or a distribution over them? As a reader I would appreciate if there's a clearly descried experiment even if it requires some of the less critical experiments need to be moved to the appendix (e.g. RQ6).

In short, I could imagine a project that studies in-context game play in games that demand common sense knowledge; with a more focused empirical section with one or two experiments that are clearly motivated and described.

[1] https://say-can.github.io/
[2] https://arxiv.org/abs/1611.02779
[2] https://proceedings.mlr.press/v162/liu22h.html

**Questions:**

1. It would be great if the authors could address comments in the weaknesses section;
2. L42-44: computing NE typically requires repeatedly computing best-responses: this is only true for Fictitious Play family of methods (e.g. PSRO) but there are many alternatives for which this is not required [1-2].
3. L45: NE is not ideal in games with more than two players: do you mean NE is difficult to compute? Could be useful if the authors could expand on this.
4. Figure 1: it's not clear to me that this figure adds to the substance of the paper and I would suggest removing Figure 1 for more details on key experiments (e.g. RQ1). This figure could also mislead as it appears that one triangle feeds into another which I don't quite see why this would be the case.
5. L74-75: AD, DPT are used for the first time (?) and yet to be defined.
6. L76-L77: "... and be used to compute NE": what can be used to compute NE? The trained model or ICE the framework? Might be worth breaking into shorter sentences for clarity.
7. L140-L150: Simplex-NeuPL [3] provides very similar motivations for not playing NE at times and could be a useful reference here as it tackles similar problem settings without using transformer or in-context learning in the LLM sense. Again Figure 2 could be removed for space or relegated to appendix. It would appear Figure 2 is stock imagery too which should be avoided? Providing a payoff table would suffice and would be more clear.
8. L182: "consists of n ..." do you mean "consists of b"?
9. L186-187: "$D_{l1}$ and $D_{lb}$ are respectively the easiest and hardest ...". This isn't true in general. If you apply PSRO to rock-paper-scissors with {90% rock, 10% paper} the initial strategy, then your second (paper) and third strategy (scissors) would both be more exploitable than your initial strategy. The convergence is in terms of meta-game NE strategy, not in terms of individual ones.
10: L196-199: "we can leverage the strategy learned via PPO...": this is not clear. Could you clarify what has been done?
11: L211: "computing the gaps would be time consuming...": what gap? Do you mean the NashConv of strategies? If so, computing the NashConv should be quite easy in these games as you would only need to compare to the BR strategy?
12: L215: It's not clear to me why you would want to mix in data random strategies at interval g. Would that lead to a jagged difficulty sequence with drops in difficulty every g steps?
13: L278: it might be clearer to avoid referring to self-play as an equilibrium finding algorithm when at L282 it's stated that "However, the self-play framework does not guarantee convergence".
14: L312: "For the out-of-distribution testbed, we randomly sampled ..." from which distribution over policies did you sample these strategies? naively the space of policy is typically exponentially large and it's not obvious to me how to create a distribution over it that ensures diversity.
15: L317: "we focus specifically on two-player games... as finding NEs in multi-player games [...] is challenging", could you clarify the NE results shown in the Kuhn-poker (3P) figure and how did you compute it?
16: L346: what does the x-axis "game iterations" mean?
17: when comparing ICE to PPO, I understood that ICE is conditioned on cross-episode trajectories that describe the interaction history. Is the PPO policy equally conditioned on cross-episode historic trajectories? If not, is each in-context episode conditionally independent from all other in-context episodes?


[1] https://proceedings.mlr.press/v119/munos20a.html
[2] https://arxiv.org/abs/2002.08456
[3] https://proceedings.mlr.press/v162/liu22h/liu22h.pdf

---

> ### Author Response · Authors · 2024-11-25
> **Reply to Reviewer XVLk (1/2)**
>
> Thank you for your valuable feedback. Below is our response to your question:
>
> ---
> **Reply to Q1**
>
> - We appreciate your thoughtful analysis of both the motivation behind our approach and the clarity of our experimental setup.
>
> - Unlike traditional uses of transformers for in-context learning, we do not leverage pre-trained general-domain knowledge. Instead, we generate domain-specific data to train a specialist transformer model capable of leveraging in-context learning. This focus on domain-specific training enables the model to address the unique challenges of multiplayer games, such as strategic reasoning and opponent exploitation, while remaining parameter-free during inference.
>
> - The key motivation for leveraging transformers lies in their ability to process long trajectories and encode complex strategic patterns within a single framework. While prior work has achieved notable results using other architectures, our results suggest that transformers, with carefully constructed domain-specific datasets, offer a flexible and scalable approach to generalization across diverse opponents. This is particularly relevant for equilibrium computation tasks, where long-term dependencies between player strategies must be modeled effectively.
>
> - As introduced in Experimental Setup. In the in-distribution testbed, the opponent policies are sampled from the same set of strategies used during the training phase. For the in-distribution testbed, we sampled about 30 opponent policies.
>
> - We use the online setting for evaluating our algorithm. Therefore, for ICE(AD), the in-context examples are generated by trajectories played with the given opponent previously, which means the context would be empty at the beginning.
>
> ---
>
> **Reply to Q2**
>
> - Here, we intend to convey that NE computation can be computationally intensive. We use PSRO as an example to illustrate why this process often involves significant computational overhead. We have revised this description in the revision.
>
> ---
>
> **Reply to Q3**
>
> - Indeed, NE is difficult to compute for games with more than two players. Additionally, in multi-player games, other equilibrium strategies may be more suitable solutions than NE, depending on the specific context and objectives of the game.
>
> ---
>
> **Reply to Q4**
>
> - Thank you for your suggestion. We have removed this figure in the revision.
>
> ---
>
> **Reply to Q5**
>
> - Thank you for pointing this out. We have revised it in the revision.
>
>
> ---
>
> **Reply to Q6**
>
> - Thank you for highlighting the ambiguity in L76-77. It should be “the trained model can be used to compute NE with equilibrium finding algorithms”. We have revised it in the revision.
>
> ---
>
> **Reply to Q7**
>
> - Thank you for your valuable feedback. We appreciate your suggestion to reference Simplex-NeuPL and have incorporated it into our discussion to provide a more comprehensive context for our motivations. In the revision, we have replaced Figure 2 with a payoff table for clarity and included Simplex-NeuPL to strengthen the connection to related work addressing similar problem settings.
>
> ---
>
> **Reply to Q8**
>
> - Thank you for pointing this out. We have revised it in the revision.
>
>
> ---
>
> **Reply to Q9**
>
> - There is a misdescription in the original paper regarding the opponent strategies generated by the learning-based method and we have revised it in the revision. These strategies are the opponent’s final strategies observed during the learning process. For the PSRO algorithm, the meta-strategy over all strategies represents the final result.
>
> - In our experiments, we use the CFR algorithm, where the opponent’s final strategies at each iteration are its average strategies among all previous strategies. As the algorithm progresses, these average strategies gradually converge toward the NE strategy. Therefore $D_{l1}$ and $D_{lb}$ are respectively the easiest and hardest strategies to exploit.
>
> ---
>
> **Reply to Q10**
>
> - The training dataset for DPT requires an optimal strategy to provide the optimal action as the label for any query state. To collect in-context data (i.e., trajectories), we use PPO to learn the optimal strategy against a fixed opponent, which is then used to generate the training trajectories. Consequently, the trained optimal strategy obtained via PPO serves as the optimal strategy required for constructing the training dataset for DPT.
>
> ---
>
> **Reply to Q11**
>
> - Yes, the gap refers to NashConv. While computing the NashConv for a single strategy is straightforward in these games, the presence of numerous opponent strategies makes the process time-consuming due to the need to compute it for all opponent strategies.

---

> ### Author Response · Authors · 2024-11-25
> **Reply to Reviewer XVLk (2/2)**
>
> ---
>
> **Reply to Q12**
>
> - The purpose of mixing in data from random strategies is to enhance the diversity of the training set, ensuring that the model can generalize to a broader range of potential opponent behaviors, including less structured or irrational strategies. We acknowledge that introducing random strategies at regular intervals may create a non-monotonic (jagged) difficulty sequence. However, our intention is not to strictly follow a monotonically increasing difficulty but rather to strike a balance between exposing the model to diverse behaviors and progressing toward harder tasks.
>
> ---
>
> **Reply to Q13**
>
> - Although current self-play does not guarantee convergence, it is often used as a framework to approximate equilibrium strategies due to its simplicity and practicality. To support this, we have added experimental results for Leduc Poker and Goofspiel, demonstrating that self-play can perform well, particularly in scenarios involving pure equilibrium strategies.
>
> ---
>
> **Reply to Q14**
>
> - Here, the “out-of-distribution” testbed refers to opponent strategies that are not included in the training dataset. To create the out-of-distribution testbed, we rerun the random generation method to produce a new set of random opponent strategies.
>
> ---
>
> **Reply to Q15**
>
> - Here, we use the CFR algorithm, as implemented in OpenSpiel, to compute the NE strategy for the three-player Kuhn Poker. While computing NEs in multi-player games is generally very challenging, three-player Kuhn Poker is relatively simple, and the CFR algorithm can effectively compute an approximate NE strategy in this case.
>
> ---
>
> **Reply to Q16**
>
> - Here, “game iterations” refers to the number of games played during the evaluation process. Each game iteration represents a complete playthrough of the game, and the x-axis tracks the cumulative number of such iterations.
>
> ---
>
> **Reply to Q17**
>
> - Indeed, ICE leverages cross-episode trajectories as part of its in-context learning process. However, while the PPO policy is not explicitly conditioned on cross-episode trajectories during inference, it does use these trajectories as training data to update the policy.
>
> - Additionally, each in-context episode is independent of all other in-context episodes, as each episode represents a single game process, and every game process is independent of one another. Therefore, we evaluate both algorithms based on the number of game iterations.
>
> ---
> We hope our response can address your concerns, and we deeply appreciate that if you could reconsider the evaluation of our paper.

---

> ### Author Response · Authors · 2024-12-02
> **Follow-Up on Review Discussion**
>
> Dear Reviewer XVLk,
>
> Thank you once again for your valuable review.
>
> As the author-reviewer discussion period is nearing its end, we would like to kindly ask if our responses have addressed your concerns. We have put a lot of effort into preparing detailed responses to your questions. We are eager to address any additional feedback or unresolved concerns you may have before the discussion period concludes.
>
> We look forward to hearing from you. Thank you for your time and consideration.
>
> Sincerely,
>
> Authors of Submission 8911

---

> ### Author Response · Authors · 2024-12-03
>
> Dear Reviewer XVLk,
>
> Thank you for your time, effort, and thoughtful feedback in reviewing our work. As the author-reviewer discussion period approaches its conclusion (11:59pm AoE on 2nd Dec), we sincerely hope that our detailed, point-by-point responses have addressed your concerns.
>
> We have dedicated significant effort to revising our paper and preparing thorough answers to your questions. If our responses have adequately resolved your concerns, we would be deeply grateful if you could reconsider your evaluation of our submission. However, if there are any remaining issues or uncertainties that you feel we have not sufficiently addressed, we kindly ask for further clarification or suggestions. We are eager to make additional improvements before the discussion period ends to ensure our work meets your expectations.
>
> Thank you again for your valuable feedback and for helping us improve our paper.
>
> Sincerely,
>
> Authors of Submission 8911

---

### Official Review · Reviewer_EYRB · 2024-11-03

**Soundness:** 3
**Presentation:** 3
**Contribution:** 2
**Rating:** 6
**Confidence:** 4

**Summary:**

This paper examines the use of in-context learning for games. The goals are to train a transformer model such that it can in-context exploit any strategy by any player of a given game. This is then used as the best-response oracle in the population-based method PSRO to approximate a Nash equilibrium in a 2P0S (two-player, zero-sum) game. It builds off of the techniques for single-player setting techniques Algorithm Distillation (AD) and Decision Pretrained Transformer (DPT).

**Strengths:**

The research direction is solid. The three goals (play as any player, exploit any strategy, and find an approximate Nash) are reasonable, and the methodology and experiments do well in moving towards these goals.

The paper is written well, the experimental results are presented nicely, and most of the methods are clear.

**Weaknesses:**

My main critique of the paper is that the three goals and the research questions are good, but they are vague enough that it's unclear what the exact hypotheses are and therefore they aren't exactly falsifiable. I think the results on "RQ3: Can ICE be used to compute NE without changing the parameters" are mixed, not positive, but the paper does not mention it as a mixed or negative result.

This is important because I think that question is the most important and is in some sense the most unambiguous. The experiment measured performed PSRO and measured how exploitability changes as the algorithm progresses. The exploitability remains quite high even at the end of the experiment, yet the paper seems to deem this an affirmative answer to the question. For this question in particular, I think an affirmative answer to the question would involve a curve which converges to 0 exploitability, or at least something lower than the value that it got. It would also be helpful to compare the exploitability of ICE to some baseline, like PSRO with RL, or double-oracle with exact BRs, and maybe also to neural fictitious self-play (NFSP).

In my opinion, claiming that ICE affirmatively answers question RQ3 is overselling the results, and I would be more inclined to give a higher score if those results were more accurately contextualized. In my opinion, the overall framing of the paper, such as in the conclusion and perhaps the abstract, should also note that the experiments show positive first steps towards the goal of RQ3, but that there is still future work needed to achieve it.

Other minor critiques:
1. The experiments for RQ3 should also be performed on Leduc and Goofspiel.
2. The setup of the Opponent Strategies Generation is ill-informed: "Intuitively, with the progress of the PSRO process, the generated opponent strategies will approach the NE strategy." That is *not* true: The opponent's *meta-strategy* will intuitively (but is also not guaranteed to) approach an NE strategy, but the best-response strategies $D_{li}$ will not. It's also not clear to me that $D_{l1}$ and $D_{lb}$ are respectively the easiest and hardest strategies to exploit. And $D_l$ is not really a diverse set of opponent strategies, since it may contain all "exploitative" (pure, or nearly pure) strategies. Because of that, they are likely all quite exploitable.
3. In the experimental results in 5.2, the BR (red line) is a fixed policy (not changing as the number of game iterations increases), right? Then it's probably clearer to take its value (either by taking its empirical value after many samples, or by simply computing its exact expected value) and plot it as a flat, horizontal line, than as a moving curve. Same with the NE (purple line).
4. It should be pointed out that ICE does not seem to converge to the BR in the OOD setting, which would presumable be a desiderata for this work.

**Questions:**

1. In Experimental Results, in RQ1 and RQ2, did you evaluate against just a single opponent per chart? Shouldn't the experiment be repeated several times, with a different opponent each time, and with error bars on the lines?
2. In RQ2: "The average returns of NE and BR strategies are not exactly zero. This deviation arises from using an approximate NE strategy as the opponent." -- Is it not expected that the average return of NE vs. NE in Kuhn poker is not 0? The Nash value is 1/18, right? [0]

[0]: https://en.wikipedia.org/wiki/Kuhn_poker

---

> ### Author Response · Authors · 2024-11-25
> **Reply to Reviewer EYRB**
>
> Thank you for your valuable feedback. Below is our response to your question:
>
> ---
> **[Weakness for RQ3]**
>
> - We agree that the current results for RQ3 are mixed, and presenting them as an affirmative answer oversells the findings.
>
> - In the revision, we have added experiments on Leduc Poker and Goofspiel (Figure 5) and clearly described the mixed nature of the results. We emphasize that while ICE demonstrates progress toward computing NE without parameter updates, the exploitability remains relatively high by the end of the experiment for games with high dynamics. However, in simpler games, such as Goofspiel with 3 cards, our ICE achieves a low exploitability. We believe this is due to the existence of a pure strategy equilibrium in this particular game.
>
> - Additionally, we have reframed the discussion to highlight these results as a promising first step rather than a definitive solution, while acknowledging the gap that remains to be addressed in future work.
> ---
> **[Opponent Strategy Generation]: The setup of the Opponent Strategies Generation is ill-informed**
> - There is a misdescription in the original paper regarding the opponent strategies generated by the learning-based method and we have revised it in the revision. These strategies are the opponent’s final strategies observed during the learning process. For the PSRO algorithm, the meta-strategy over all strategies represents the final result.
>
> - In our experiments, we use the CFR algorithm, where the opponent’s final strategies at each iteration are its average strategies among all previous strategies. As the algorithm progresses, these average strategies gradually converge toward the NE strategy. Therefore $D_{l1}$ and $D_{lb}$ are respectively the easiest and hardest strategies to exploit.
>
> ---
>
> **[Experimental Results in 5.2]: The BR (red line) is a fixed policy (not changing as the number of game iterations increases), right?**
>
> - BR (red line) and NE (purple line) are fixed policies. As our results are obtained from simulations, we plotted the curve to illustrate that the estimation for these policies converges to a fixed value as the number of simulation iterations increases. We appreciate your suggestion, and we will consider representing these values as flat horizontal lines in revisions to make the results clearer and emphasize the fixed nature of these policies.
> ---
> **[OOD setting]: It should be pointed out that ICE does not seem to converge to the BR in the OOD setting.**
>
> - Thanks for pointing this out. We have revised it in our revision.
>
> ---
> **Q1:  Did you evaluate against just a single opponent per chart?**
> - The experimental results are based on evaluations against a set of opponents, as outlined in the Experimental Setup.  Specifically, we used 30 opponents for the in-distribution testbed and 20 opponents for the out-of-distribution testbed.  Since these opponents exhibit significant variability and the utility against them has a high variance—ranging from the lowest to the highest returns—we opted to plot only the average results in the charts without including error bars.
>
> ---
> **Q2: Is it not expected that the average return of NE vs. NE in Kuhn poker is not 0?**
>
> - Thank you for the comment. The Nash value for Kuhn Poker is indeed 1/18. Here, we intend to highlight that the sum of the average returns of NE and BR strategies for all players is not exactly zero due to the approximation of NE in our experiments. This has been clarified in the revised manuscript.
>
> ---
> We hope our response can address your concerns, and we deeply appreciate that if you could reconsider the evaluation of our paper.

---

> > ### Comment · Reviewer_EYRB · 2024-11-26
> >
> > Thanks for engaging with my review!
> >
> > > Additionally, we have reframed the discussion to highlight these results as a promising first step rather than a definitive solution, while acknowledging the gap that remains to be addressed in future work.
> >
> > Nice, this resolves my biggest complaint with the paper. I have raised my review score.
> >
> > > There is a misdescription in the original paper regarding the opponent strategies generated by the learning-based method and we have revised it in the revision. These strategies are the opponent’s final strategies observed during the learning process. For the PSRO algorithm, the meta-strategy over all strategies represents the final result.
> >
> > Interesting -- this makes more sense.
> >
> > > The experimental results are based on evaluations against a set of opponents, as outlined in the Experimental Setup. Specifically, we used 30 opponents for the in-distribution testbed and 20 opponents for the out-of-distribution testbed. Since these opponents exhibit significant variability and the utility against them has a high variance—ranging from the lowest to the highest returns—we opted to plot only the average results in the charts without including error bars.
> >
> > If the results exhibit high variance, shouldn't the plots include error bars to indicate that? Otherwise by looking at the results, it's impossible to tell what kind of variance there is, and a reader will assume everything is signal and not noise.

---

> > > ### Author Response · Authors · 2024-11-27
> > >
> > > Thank you for your reply and thoughtful suggestions regarding the presentation of the experimental results. We have revised all the figures in the updated version of the paper to address your concerns.
> > >
> > > - We have added error shadows to the plots to illustrate the variance in the results.
> > > - We have updated the BR (red line) and NE (purple line) to represent flat, horizontal lines, reflecting their fixed values.
> > >
> > > We hope these changes address your concerns and improve the clarity and accuracy of the figures. Please let us know if you have further suggestions or feedback.

---

### Official Review · Reviewer_Xvmq · 2024-11-04

**Soundness:** 3
**Presentation:** 3
**Contribution:** 2
**Rating:** 6
**Confidence:** 4

**Summary:**

The goal of this work is to leverage in-context learning through the use of a transformer agent for multi-agent systems such that this single model can adjust to playing as any player and exploiting any type of opponent entirely through the context within its input and not through parameter fine-tuning.

**Strengths:**

- I think the approach and the framework is interesting, and the notion of in-context learning as a method of fast adaptation to any player strategy is interesting in the field of game solving.
- I think the paper is well-written and generally well-explained.
- The empirical results, whilst maybe lacking in terms of complexity of game explored, are encouraging for the proposed framework

**Weaknesses:**

- In terms of opponent strategies generation, I think it is missing some nuance. In particular, I am not sure that either the random generation or the learning-based generation necessarily produce a meaningful & diverse dataset of opponent strategies. Whilst random generation can guarantee a form randomness, I am not sure that it would produce relevant behaviour to actually *learn* against (i.e. if we consider from the perspective that this approach should help with dealing with a wide set of opponents, it is unlikely that a real-world opponent would perform entirely at random). Furthermore, vanilla PSRO has been demonstrated in many papers to not generate a population of diverse agents and that is why there is a long line of literature looking at this exact problem: e.g. see those that cite 1) Open-Ended Learning in Symmetric Zero-Sum Games (Balduzzi et al. 2019) or 2) Modelling Behavioural Diversity for Learning in Open-Ended Games (Perez-Nieves et al. 2021). Therefore, can the authors demonstrate the diversity of the data that is collected?
- For the learning-based generated opponent strategies, it would also be good for the authors to discuss / demonstrate why they have called the final strategies added to the population as the most difficult task? I would be more inclined to say that the NE distribution over the *final* population is the most difficult task, but not the final strategy itself (e.g. it may only constitute a small part of the true NE support). Could the authors maybe present a stronger argument for why the order of teh curriculum they proposed is good for this framework?
- To me personally, it feels counterproductive to the learning process to include the random strategies to the curriculum process at all (they will most likely be such poor strategies that they could interrupt the curriculum as they are possibly so different - the authors also mention the catastrophic forgetting problem and I think the influence of these random strategies could emphasise this).
- My general concern with the framework is that, whilst the results do suggest that in-context learning can allow the model to perform well in comparison to other training approaches, is it truly beneficial if we still need to do the pre-training phase (which requires e.g. generation of PSRO trajectories and therefore a large amount of e.g. RL training anyway)? I think the authors should spend more time discussing why this is an approach that could be ultimately be better even with the potentially large amounts of pre-training required?
- I think RQ5 could use a small ablation in terms of which part of the CL is used (e.g. CL with PSRO generated strategies only, random only, both etc...)

**Questions:**

- I don't understand fully what is meant to by the third part of the desiderata 'used to compute the NE without changing the parameters' - especially in the context of the motivating example. The motivating example seems to suggest that we want an algorithm that is not *necessarily* searching for the NE given the behaviour of the opponent player. This seems counter-intuitive to the third desiderata, or I may be misinterpreting this point.
- In the card games, what does it mean for the framework to handle different players? Are the different players just defined by order of which of the players plays first?
- For the out-of-distribution testbed, where were the randomly sampled opponent strategies from? Were they just random strategies?
- In Fig. 10, what is the difference between the two subplots in (a) and (b)?

---

> ### Author Response · Authors · 2024-11-25
> **Reply to Reviewer Xvmq (1/2)**
>
> Thank you for your valuable feedback. Below is our response to your question:
>
> ---
> **Q1: Can the authors demonstrate the diversity of the data that is collected?**
>
> - Our generated opponent strategies include both random strategies and those derived from the equilibrium finding algorithm. As outlined in the paper, the random generation step serves to create a baseline of varied and unpredictable strategies, mimicking scenarios where opponents may act irrationally. These random strategies inherently contribute to the overall diversity of the dataset.
>
> - We acknowledge the limitations in the diversity of strategies generated by the PSRO algorithm, as highlighted in the literature. In the paper, we use the PSRO algorithm as an illustrative example to demonstrate how opponent strategies can be recorded (i.e., the meta-strategy over strategy pools) during the equilibrium learning process, capturing different levels of opponent capabilities. In our experiments, we apply the CFR algorithm to generate these opponents. Specifically, the average strategies generated among all previous strategies during each iteration of CFR are added to the opponent strategy pool, ensuring a broad representation of different skill levels and behavior types. These strategies are designed to reflect a spectrum of opponent capabilities, with their varying proximity to the NE strategy. We also acknowledge a misdescription in the original paper and have corrected it in the revised version.
>
> - By combining these two types of opponent strategies—randomly generated and learning-based—we aim to construct a dataset that is diverse both in behavioral patterns and skill levels.
>
> ---
> **Q2: Could the authors maybe present a stronger argument for why the order of the curriculum they proposed is good for this framework?**
>
> - There is a misdescription in the original paper regarding the opponent strategies generated by the learning-based method and we have revised it in the revision. These strategies are the opponent's final strategies observed during the learning process. For the PSRO algorithm, the meta-strategy over all strategies represents the final result.
>
> - In our experiments, we use the CFR algorithm, where the opponent’s final strategies at each iteration are its average strategy among all previous strategies. As the algorithm progresses, these average strategies gradually converge toward the NE strategy. If we consider the NE strategy as the most challenging opponent to exploit, the sequential order of these strategies naturally forms a suitable curriculum for training.
> ---
> **Q3: To me personally, it feels counterproductive to the learning process to include the random strategies in the curriculum process at all**
> - As we introduced, the random generation of opponent strategies is intended to enhance the overall diversity of the opponent strategy set. While learning-based strategies provide structured and progressively refined behaviors, they may lack sufficient diversity to fully train a robust and adaptable model. By incorporating random strategies into the curriculum, we aim to expose the model to a broader spectrum of behaviors, including unpredictable and irrational actions, thereby enhancing its robustness.
>
> - To address the concern about catastrophic forgetting, our curriculum learning framework includes a revisiting mechanism. This mechanism allows the model to periodically retrain on previously encountered strategies, ensuring that knowledge gained from earlier tasks is retained while the model learns to adapt to new strategies. This approach mitigates the potential for random strategies to disrupt the learning process, maintaining stability and preventing the model from overfitting to specific opponent behaviors.
>
> ---
> **Q4: Why this is an approach that could be ultimately be better even with the potentially large amounts of pre-training required?**
>
> - *Parameter-Free.* While it may appear resource-intensive, once pre-trained, the model can adapt to new scenarios or opponents through in-context examples without requiring any further updates to its parameters. This is particularly advantageous in real-world applications where retraining or fine-tuning may be infeasible due to time, computational constraints, or the need for rapid deployment.
>
> - *Generalizability.* The model trained through our approach can handle a wide variety of opponents, ranging from random strategies to those close to Nash Equilibrium, providing robustness and flexibility in unpredictable environments.
>
> - *Efficiency During Inference.* While the pre-training phase may involve significant computation, the inference process is lightweight, as it leverages a pre-trained model and in-context learning to dynamically adapt. This makes it suitable for scenarios where inference needs to be conducted efficiently and repeatedly.

---

> ### Author Response · Authors · 2024-11-25
> **Reply to Reviewer Xvmq (2/2)**
>
> ---
> **Q5: Ablation in terms of which part of the CL is used**
>
> - We have updated Figure 7 in the revision to include two additional ablation results for a fair comparison. All comparisons are conducted using the same set of opponent strategies. Specifically, we compare our method with two training strategies: Training first on random opponents, followed by learning-based opponents (“Random First”), and Training first on learning-based opponents, followed by random opponents (“Learned First”).
>
> - As shown in Figure 7, ICE, Random First, and Learned First all outperform ICE without CL. Interestingly, ICE, Random First, and Learned First achieve similar overall performance, which may be attributed to the inherent curriculum-like nature of learning-based opponents.
>
> - However, we observe that Learned First performs better in in-distribution cases, likely due to the stronger initial focus on structured strategies and Random First performs better in out-of-distribution cases, benefiting from early exposure to diverse random strategies. Our ICE method demonstrates more stable performance across both in-distribution and out-of-distribution settings. This indicates that integrating random opponents into training with learning-based opponents enhances the robustness of the training process, leading to a more balanced and adaptive model.
>
> ---
> **Q6: About the third part of the desiderata**
>
> - The third desideratum refers to the ability of the pre-trained model to assist in computing Nash Equilibrium (NE) strategies, without requiring additional parameter updates. Specifically, the model leverages its in-context learning capabilities to represent and adapt strategies dynamically by modifying the context, rather than altering its internal parameters.
>
> - The motivating example highlights the limitations of NE strategies in certain scenarios, particularly against opponents who deviate from equilibrium behavior. It demonstrates the need for a model that can exploit such deviations to maximize utility, rather than rigidly adhering to NE strategies.
>
> - The third desideratum and the motivating example serve complementary purposes. The motivating example illustrates the flexibility of the proposed framework in exploiting non-equilibrium opponents, while the third desideratum ensures that the same model can also contribute to NE computation when required.
>
> ---
> **Q7: In the card games, what does it mean for the framework to handle different players? Are the different players just defined by order of which of the players plays first?**
>
> - Note that these card games are not symmetric games. “different players” refers to the distinct roles that each player assumes in the game, not just the order in which they play. For example, the information available to a player at any point in the game varies depending on their role and each player may face different sets of possible actions depending on the game state and their position in the game.
>
> - Optimal strategies for one player often depend on the strategies of the other players, which makes playing as different players a distinct task. Our framework is designed to train a single model that can play as any player in the game. This means the model can adapt its behavior based on the player role it is assigned and the corresponding context. Therefore, handling different players in our framework involves adapting to the unique roles and decision-making scenarios of each player, which go beyond just the order of play.
>
>
> ---
> **Q8: For the out-of-distribution testbed, where were the randomly sampled opponent strategies from? Were they just random strategies?**
>
> - To generate the out-of-distribution testbed, we rerun the random generation method to generate these random opponents not just random strategies.
>
> ---
> **Q9: In Fig. 10, what is the difference between the two subplots in (a) and (b)?**
>
> - The two subplots in (a) and (b) represent results for in-distribution and out-of-distribution test cases, respectively. We have clarified this distinction in the revised manuscript to ensure a better understanding.
>
> ---
> We hope our response can address your concerns, and we deeply appreciate that if you could reconsider the evaluation of our paper.

---

> ### Author Response · Authors · 2024-12-02
> **Follow-Up on Review Discussion**
>
> Dear Reviewer Xvmq,
>
> Thank you once again for your valuable review.
>
> As the author-reviewer discussion period is nearing its end, we would like to kindly ask if our responses have addressed your concerns. We have put a lot of effort into preparing detailed responses to your questions. We are eager to address any additional feedback or unresolved concerns you may have before the discussion period concludes.
>
> We look forward to hearing from you. Thank you for your time and consideration.
>
> Sincerely,
>
> Authors of Submission 8911

---

### Author Response · Authors · 2024-11-25

Dear Reviewers,

We sincerely appreciate your time and effort in reviewing our work, as well as your valuable and constructive comments.

We have carefully addressed all the questions and concerns you raised and uploaded a revised version of our paper. Below, we summarize some of the key modifications:
- We added experimental results on Leduc and Goofspiel games to further evaluate the computation of NE strategies.
- We included additional ablation studies focused on curriculum learning (CL) to provide deeper insights into its effects.
- We have corrected all typos and misdescriptions throughout the paper for improved clarity and precision.

We hope that our responses and revisions have adequately addressed your concerns. If so, we respectfully request that you consider raising your score. We are also open to further discussions or clarifications should you have any remaining questions or suggestions.

Once again, we sincerely thank the reviewers for their efforts and valuable feedback.

Sincerely,

Authors of Submission 8911

---

### Meta-Review · Area_Chair_GNhU · 2024-12-11

**Metareview:**

This paper proposes to extend the in-context-learning (ICL) properties of Transformers to multi-agent settings, such that each agent can fast-adapt to playing any role and against any opponents entirely through context. The introduction of ICL into game-playing is relatively new, and the problem is well-motivated, with good research questions being asked. The paper is well-written and easy to follow generally. However, there were some concerns regarding the generation of the opponents/data, some algorithm design choices, the actual advantage of ICL due to the requirement of pre-training, and the overselling of some results. The clarity and rigor of the exposition and the sufficiency of the experiments could also have been strengthened. The authors did a good job clarifying some misunderstandings/misdescriptions in the revised version, but the paper is still borderline in terms of the readiness for publication. Given the recent breakthroughs in the theoretical understanding of ICL of transformers in single-agent settings, the paper could have also been strengthened with some analyses, even in canonical settings, to complement the experimental results. I suggest the authors incorporate the feedback this round in preparing the next version of the paper.

**Additional Comments On Reviewer Discussion:**

There were some concerns regarding the misdescriptions of some algorithm details and experimental results, and have been addressed by the authors' rebuttal and revision. However, the concerns regarding the sufficiency of the experiments, especially if the claims are all well supported by the experiments, were not fully addressed.

---

### Decision · Program_Chairs · 2025-01-22

Reject